# Frequency-Domain Better than Time-Domain for Causal Structure Recovery in Dynamical Systems on Networks

**Mohammed Tuhin Rana**
University of Minnesota
rana0082@umn.edu

**Mishfad Shaikh Veedu**
Google LLC
mishfad@google.com

**James Melbourne**
Centro de Investigación en Matemáticas
james.melbourne@cimat.mx

**Murti V. Salapaka**
University of Minnesota
murtis@umn.edu

## Abstract

Learning causal effects from data is a fundamental and well-studied problem across science, especially when the cause-effect relationship is static in nature. However, causal effect is less explored when there are dynamical dependencies, i.e., when dependencies exist between entities across time. In general, it is not possible to reconstruct the causal graph from data alone. The conventional static causal structure recovery algorithms employ tests such as the Fischer-z test and the chi-square test to assess the conditional independence (CI) of data which forms the basis for recovering Markov Equivalent Graphs (MEGs) wherein causal structure can be recovered partially. For data that are dynamically related, multivariate least square estimation, based on Wiener Filters (WFs) relying on second order statistics for estimating a data stream from other streams, provides a means of recovering influence structures of the directed network underlying the data. Here, WF based projections can be determined in time-domain or in frequency-domain; the question this article sets out to answer is which is better? Here, we obtain concentration bounds on the accuracy of the WF estimation in both time and frequency-based approaches. Exploiting the computation speed of Fast Fourier Transform (FFT), we establish that the frequency domain provides distinct advantages. Moreover, frequency domain projections involve complex numbers; we establish that the phase properties of the resulting estimates can be effectively leveraged for better recovery of the MEG in a large class of networks; the time-domain has no analogue of phase. Thus we report the "Wiener-Phase" algorithm provides the best accuracy as well as computational advantages. We validate the theoretical analysis with numerical results. Performance comparison with state of the art algorithms are also provided. Further, the proposed algorithms are validated on a real field dataset known as the "river-runoff" dataset collected from the online repository of CauseMe, and on measurement data from transistor based circuits.

## 1 Introduction and Literature Survey

Causal identification from data is an active and important research area relevant to multiple domains including climate science Pérez-Suay & Camps-Valls (2018), economics Carfi & Caristi (2008), neuroscience Ramirez-Villegas et al. (2021), and biology Hu et al. (2018) Lu et al. (2023). There is considerable prior art on causation especially when the interactions are static in nature (see Pearl et al. (2016), Peters et al. (2017), Spirtes et al. (2000), Wren et al. (2022) and the references therein). In the cases where the entities are modeled as random variables it is not possible in general to reconstruct the causal graph from observational data alone. Indeed, Markov equivalent graphs (MEGs) that capture the same set of conditional independence (CI) relations can be determined from data Spirtes et al. (2000). Some recent works on recovering a unique causal graph when the underlying data generative system is assumed to have more structure are presented in Peters et al. (2013), Shimizu et al. (2006).

Causal inference is more challenging in the presence of dynamical (across time) dependencies Costanzo & Yağan (2020); Krishnan et al. (2023); Peters et al. (2013). One of the established techniques for identifying causation in systems with dynamics is Granger causality Granger (1969), Bennett & Yu (2020), which leverages temporal structure assuming delays are present in interactions. Such an assumption is rendered problematic when the data is collected at slower rates than the time-constants at which the dynamics evolve, thus precluding a number of practical scenarios. Another approach in handling dependencies across times is to consider the series at every time instant as a random variable thereby mapping the problem to a static version. The difficulties with such an approach stem from the lack of information on the size of the horizon in the past and future to be considered and the combinatorial explosion of the number of variables that entail Ghahramani (2006), Lohmann et al. (2012). In a class of approaches for unveiling dynamic dependencies, models of how the data is being generated is assumed and the causal graph structure is recovered via an estimation of model parameters. Examples of such models include vector auto regressive (VAR) models Krishnan et al. (2023), Peters et al. (2013), additive noise models (ANM) Costanzo & Yağan (2020), Peters et al. (2013) and neural network-based models (NNM) Moraffah et al. (2021). In another recent line of work, the main gist of the approach is to exploit the asymmetry in the relationships of a dependency and the inverse of the dependency. These asymmetries can result from nonlinear maps or from linear filters with inverses that have discernible differences that, for example, are characterizable via power spectral densities Besserve et al. (2022), Shajarisales et al. (2015). Here, it is to be noted that works such as Besserve et al. (2022) and Shajarisales et al. (2015) try to infer cause and effect relationship between two variables; however, the causal graph structure recovery entails determining how the influence flows from a variable to another with the possibility of intermediate variables between the cause and the effected variables.

Compared to the case when interactions are static, the literature on causal structure recovery for systems with dynamic interactions is sparse. Several recent works have bridged the causality literature (Spirtes et al. (2000)) with multivariate projections of dynamically related time-series data Costanzo & Yağan (2020), Materassi & Salapaka (2016), Materassi & Salapaka (2019). It was shown in Materassi & Salapaka (2012) that the Wiener Filter (WF)-based projections can be employed to recover the moral graph of entities interacting via linear time-invariant dynamics. Materassi & Salapaka (2013) proposed that the WF-based tests can in principle be employed for performing CI tests in linear dynamical systems. We emphasize that effective and implementable methodologies for causal inference in linear dynamical systems remain much less developed than their static dependency counterparts. Moreover, WF based projections are often determined in the time-domain; as we demonstrate TD based approaches can be computationally burdensome.

This article develops a framework to reconstruct the MEG for a system with dynamical interactions using a frequency domain (FD) approach. The main contributions are: 1) It is shown with examples that the WF-based reconstruction techniques provide superior performance in systems with linear dynamical interaction when compared to the conventional static counterpart. For the implementation of causation algorithms such as PC and FCI Spirtes et al. (2000). WF-based CI tests are provided. 2) To reduce the computational complexity of the WF implementation, an FFT-based approach is proposed which is proved both theoretically and in simulations to have faster run times than TD based approaches. 3) The pairwise CI test-based algorithms such as PC is combinatorial in nature, which is intractable in bigger networks. To tackle this issue a computationally attractive algorithm called Wiener-Phase is proposed to reconstruct the MEG, albeit under some assumptions that is applicable in systems such as power-grid, consensus dynamics, and chemical reactions. 4) For estimating Wiener coefficients from time-series data, a non-asymptotic concentration bound and a sample complexity analysis are provided for both the time and FFT-based approaches. The results show that the FFT-based approach is computationally superior to time-based approach by a factor of $O(L^2/\log N)$, where $L$ is the number of future and past samples used for WF TD estimation and $N$ is the FFT window length. The simulation results corroborate the conclusion that the FFT approach gives better reconstruction error as well.

Notations: $\mathbb{R}, \mathbb{C}, \mathbb{Z}, \mathbb{N}$ denote the set of real numbers, complex numbers, integers and natural numbers respectively; $[N] := \{0, \ldots, N-1\}$; For any time signal, $x := \{x(t) : t \in \mathbb{Z}\}$, $X$ denotes the Fourier transform of $x$, $X(\omega) := \sum_{n=-\infty}^{\infty} x(n)e^{-j\omega n}$; for a stochastic processes, $x$, $\Phi_{xx}(\omega)$ denotes the power spectral density (PSD) of $x$; for a complex number, $x = x_R + ix_I$, $\angle x$ denotes $\arctan(x_I/x_R)$; $\Omega$ denotes the interval $[0, 2\pi]$ and $\Omega_N$ denotes $\{0, 2\pi/N, \ldots, 2\pi(N-1)/N\}$; for a transfer function, $X, X \neq 0$ means $X$ is not identically zero. In a given directed graph, $Pa(i), Ch(i), Sp(i)$ denotes

the set of parents, children, and spouses of node $i$ respectively. $\ell^1$ denotes the space of absolutely summable sequences.

## 2 PRELIMINARIES AND RESULTS ON WIENER FILTER COMPUTATIONS

### 2.1 LINEAR DYNAMIC INFLUENCE MODELS (LDIM)

Consider a network with $n$ nodes, each node $i \in \{1, \dots, n\}$ having the time-series measurements, $x_i := \{x_i(k)\}_{k \in \mathbb{Z}}$, governed by the convolutional model,

$$\mathbf{x}(k) = \sum_{l=-\infty}^{\infty} \mathbf{h}(l)\mathbf{x}(k-l) + \mathbf{e}(k), \tag{1}$$

where $\mathbf{x}(k) := [x_1(k) \ \dots \ x_n(k)]^\top \in \mathbb{R}^n$, $\mathbf{h}(\ell) \in \mathbb{R}^{n \times n}$, and $\mathbf{e}(k) := [e_1(k) \ \dots \ e_n(k)]^\top \in \mathbb{R}^n$ is the vector of exogenous noise sources with $e_i, e_j$ jointly wide sense stationary. That is, for every $i, j \in \{1, \dots, n\}$, there exists an $\alpha_i \in \mathbb{R}$ such that $\mathbb{E}[e_i(k)] = \alpha_i$ for every $k$, and $\mathbb{E}[e_i(k)e_j(\ell)] = \mathbb{E}[e_i(k-\ell)e_j(0)] =: R_{ij}(k-\ell)$. Taking Discrete Time Fourier Transform (DTFT), equation 1 can be represented using the following linear dynamical influence model (LDIM),

$$\mathbf{X}(\omega) = \mathbf{H}(\omega)\mathbf{X}(\omega) + \mathbf{E}(\omega), \ \forall \omega \in \Omega, \tag{2}$$

where $\mathbf{X}(\omega) = [X_1(\omega) \ \dots \ X_n(\omega)]^\top \in \mathbb{C}^n$, and $\mathbf{E}(\omega) = [\mathbf{E}_1(\omega) \ \dots \ \mathbf{E}_n(\omega)]^\top \in \mathbb{C}^n$. $\mathbf{H}(\omega) \in \mathbb{C}^{n \times n}$ is a well posed transfer function, $(\mathbf{I} - \mathbf{H}(\omega))$ is invertible as well as every entry of $(\mathbf{I} - \mathbf{H}(\omega))^{-1}$ is analytic, and $\mathbf{H}_{ii}(\omega) = 0$ for every $\omega \in \Omega$. Moreover, it is assumed that the power spectral density matrix (PSDM) of $\mathbf{e}$, $\Phi_{\mathbf{E}}(\omega) := \sum_{k \in \mathbb{Z}} R_{\mathbf{ee}}(k)e^{-j\omega k}$, is positive definite and diagonal for every $\omega \in \Omega$.

### 2.2 GRAPH DEFINITIONS

The structure induced by LDIM equation 2 can be represented using a graph $G = (V, \vec{\mathcal{E}})$, where $V = \{1, \dots, n\}$ and $\vec{\mathcal{E}} = \{(u,v) : H_{vu} \neq 0\}$, that is, there exists an edge $u \to v$ in $\vec{\mathcal{E}}$ if $\mathbf{H}_{vu}(\omega) \neq 0$ for some $\omega \in \Omega$. If $u \to v$ exists in $\vec{\mathcal{E}}$ then $u$ is called parent of $v$ ($u \in Pa(v)$) and $v$ is called child of $u$ ($v \in Ch(u)$). If $u \to i \leftarrow v$ exists in $\vec{\mathcal{E}}$ then $u$ and $v$ are spouses ($u \in Sp(v)$). If $u \in Sp(v)$ but $u \notin Pa(v) \cup Ch(v)$ then $u$ is a strict spouse of $v$. $u$ is said to be a kin of $v$ in $G$, denoted, $u \in kin_G(v)$, if at least one of the following exist in $\vec{\mathcal{E}}$: $u \to v$, $v \to u$, $u \to i \leftarrow v$ for some $i \in V$. $kin_G(j)$ denotes the set of kins of $j$ in $G$. A *chain* from node $i$ to node $j$ is an ordered sequence of edges in $\vec{\mathcal{E}}$, $((\ell_0, \ell_1), (\ell_1, \ell_2) \dots, (\ell_{n-1}, \ell_n))$, where $\ell_0 = i$, $\ell_n = j$, and $(\ell_k, \ell_{k+1}) \in \vec{\mathcal{E}}$. Skeleton (topology), $Skel(G)$, of a directed graph, $G = (V, \vec{\mathcal{E}})$, is an undirected graph; $Skel(G) := (V, \mathcal{E})$ where an edge $(u, v) \in \mathcal{E}$ if either $(u, v) \in \vec{\mathcal{E}}$ or $(v, u) \in \vec{\mathcal{E}}$. A *path* from node $i$ to node $j$ in the directed graph $G$ is an ordered set of edges $((\ell_0, \ell_1), (\ell_1, \ell_2) \dots, (\ell_{n-1}, \ell_n))$ with $\ell_0 = i$, $\ell_n = j$ in its skeleton, $(V, \mathcal{E})$, where $\{(\ell_k, \ell_{k+1})\} \in \mathcal{E}$. A path of the form $((\ell_0, \ell_1), \dots, (\ell_{n-1}, \ell_n))$ has a collider at $\ell_k$ if $\ell_{k-1} \to \ell_k \leftarrow \ell_{k+1}$ exists in $G$. Consider disjoint sets $X, Y, Z \subset V$ in a directed graph $G = (V, \vec{\mathcal{E}})$. Then, $X$ and $Y$ are *d-separated* given $Z$ in $G$, denoted d-sep$_G(X, Z, Y)$ if and only if every path between $x \in X$ and $y \in Y$ satisfies at least one of the following: 1) The path contains a non-collider node $z \in Z$. 2) The path contains a collider node $w$ such that neither $w$ nor the descendants of $w$ are present in $Z$. In-nodes of $i$ are given by the set $\{j : (i, j) \in \vec{\mathcal{E}}\}$.

### 2.3 WIENER FILTER-BASED GRAPH/SKELETON LEARNING

Here we first present a method for estimating the $i^{th}$ time-series from another set of time-series. Let $x := \{x(t) : t \in \mathbb{Z}\}$ and $y := \{y(t) : t \in \mathbb{Z}\}$ be $p$ and $q$ dimensional random processes with $x(t) = [x_1(t) \ x_2(t) \ \cdots \ x_p(t)]^\top$ and $y(t) = [y_1(t) \ y_2(t) \ \cdots \ y_q(t)]^\top$. Consider the problem of finding the minimum mean square error (MMSE) estimate of $y$ given $x$, defined by:

$$\widehat{y} := \arg \min_{\substack{\tilde{y} := [\tilde{y}_1(n) \ \cdots \ \tilde{y}_q(n)]^\top \\ \tilde{y}_j \in \mathcal{M}, \ j=1,\dots,q}} \mathbb{E}\left[\|y(n) - \tilde{y}(n)\|_2^2\right], \ n \in \mathbb{Z}, \tag{3}$$

where $\mathcal{M} := span\{x_j(k) : j = 1, \dots, p, -\infty < k < \infty\}$. The solution, $\widehat{y}(n) := [\widehat{y}_1(n) \ \widehat{y}_2(n) \ \cdots \ \widehat{y}_q(n)]^\top$ is given by the Wiener filter (WF) Kailath et al. (2000), described by: $\widehat{y}_i(n) = \sum_{k=-\infty}^{\infty} h_i(k)x(n-k)$, $i = 1, \dots, q$, where $h_i(k) \in \mathbb{R}^p$ are the WF coefficients, with $h_i(k) = \frac{1}{2\pi} \int_0^{2\pi} \Phi_{y_i x}(\omega)\Phi_{xx}^{-1}(\omega)e^{j\omega k}d\omega$; $\Phi_{y_i x}(\omega) := \sum_{k=-\infty}^{\infty} R_{y_i x}(k)e^{-j\omega k}$, $R_{y_i x}(k) := \mathbb{E}[y_i(k)x^\top(0)]$. Alternately, the optimal solution is described in the FD by $\widehat{Y}(\omega) = $

$\Phi_{yx}(\omega)\Phi_{xx}^{-1}(\omega)X(\omega), \forall\, \omega \in \Omega$, where $\widehat{Y}(\omega)$ is the Fourier transform of $\widehat{y}$. The solution is unique if $\Phi_{xx}(\omega)$ is positive definite for every $\omega \in \Omega$. In the above description if we let $C \subseteq V \setminus \{i\}$ and instantiate $y = x_i$ and $x = x_C$, then the MMSE estimator, $\widehat{x}_i$, of the process $x_i$, given the time-series $x_C$ satisfies $\widehat{X}_i(\omega) = W_{i \cdot C}(\omega)X_C(\omega)$ where $W_{i \cdot C}(\omega) = \Phi_{x_i x_C}(\omega)\Phi_{x_C x_C}^{-1}(\omega)$. Here $W_{i \cdot C}(\omega)$ is the multivariate WF obtained while projecting the time-series $x_i$ onto $x_C$ Kailath et al. (2000). The entry of $W_{i \cdot C}(\omega)$ corresponding to $j^{th}$ time-series in the set $C$ is denoted by $W_{i \cdot C}[j](\omega)$.

## 2.4 TD approach to determining WF

Next, we briefly outline the conventional *TD-based approach* of estimating the multivariate WF coefficients from finite data. Given the time-series data $\{\mathbf{x}(t)\}_{t=0}^{T}$, consider the projection of $x_i(t)$ to the present, the past $L$ values, and the future $L$ values of $x_j(t)$, $j \in C \subseteq V \setminus \{i\}$ with $m := |C|$. That is, for every $t = L, \ldots, T - L$, project $x_i(t)$ to $span\{x_j(k) : t - L \le k \le t + L, j \in C\}$. A TD estimate of the WF, $\mathbf{w}_{i \cdot C}^{(T,L)} \in \mathbb{R}^{(2L+1) \times m}$ can be computed using the following least square formulation,

$$\beta^* := \arg \min_{\beta \in \mathbb{R}^{m(2L+1)}} \frac{1}{T+1}\|\mathbf{x}_i - \mathbf{y}_C\beta\|_2^2, \tag{4}$$

where $\mathbf{x}_i := [x_i(T-L)\ x_i(T-L-1)\ \ldots\ x_i(L)]^\top$ and $\mathbf{y}_c$ is given by

$$\mathbf{y}_C := \begin{bmatrix} x_{c_1}(T) & x_{c_1}(T-1) & \cdots & x_{c_1}(T-2L) & \cdots & x_{c_m}(T) & \cdots & x_{c_m}(T-2L) \\ x_{c_1}(T-1) & x_{c_1}(T-2) & \cdots & x_{c_1}(T-2L-1) & \cdots & x_{c_m}(T-1) & \cdots & x_{c_m}(T-2L-1) \\ \vdots & & & & & & & \\ x_{c_1}(2L) & x_{c_1}(2L-1) & \cdots & x_{c_1}(0) & \cdots & x_{c_m}(2L) & \cdots & x_{c_m}(0) \end{bmatrix}. \tag{5}$$

The WF estimate, $\mathbf{w}_{i \cdot C}^{(T,L)}$, is obtained by reshaping $\beta^*$ into a matrix of the dimension $(2L+1) \times m$, where the $i$-th $(2L+1)$ entries correspond to the coefficient of $c_i$. Without loss of generality, the rows of $\mathbf{w}_{i \cdot C}^{(T,L)}$ are indexed from $-L$ to $L$ and the columns from $1$ to $m$. The Fourier transform of $\mathbf{w}_{i \cdot C}^{(T,L)}$, $\mathbf{W}_{i \cdot C}^{(T,L)}(\omega) \in \mathbb{C}^m$ is $\mathbf{W}_{i \cdot C}^{(T,L)}(\omega) := \sum_{k=-L}^{L} \mathbf{w}_{i \cdot C}^{(T,L)}(k)e^{-j\omega k}$, where $\mathbf{w}_{i \cdot C}^{(T,L)}(k)$ is $k$-th row of $\mathbf{w}_{i \cdot C}^{(T,L)}$. The sample complexity of estimating $\mathbf{W}_{i \cdot C}^{(T,L)}(\omega)$ using equation 4 scales as $T = O\left(Ln\right)$ as provided in Theorem 5.1.

## 2.5 FFT-based Computation of WF

Another approach for determining the WF coefficients is to first transform every time-series to its Fast Fourier Transform (FFT) representation, followed by *projections in the FD*. This approach involves sampling $\Omega = [0, 2\pi]$ at $N$, equally spaced points, $\Omega_N = \{\omega_0, \ldots, \omega_{N-1}\}$, where $\omega_k := \frac{2\pi k}{N}$ with $N = 2^a$, $a \in \mathbb{N}$, as shown in Appendix D. This process of transforming the time-series to FFT representation is as follows. Consider the $i^{th}$ time-series $x_i$ which is partitioned to $R$ segments, each of length $N$, with the $r^{th}$ segment denoted by $x_i^r := \{x_i((r-1)N),\ x_i((r-1)N+1), \ldots, x_i((r-1)N+N-1)\}$. Thus, the time-series $x_i$ is given by $\{(x_i^r(t))_{t=0}^{N-1}\}_{r=1}^{R}$, where $x_i^r(t) := x_i((r-1)N+t)$. Using the $r^{th}$ segment of the $x_i$ trajectory, given by $x_i^r(0), \ldots, x_i^r(N-1)$, the FFT, $\mathbf{X}_i^r(\omega_k) = \frac{1}{\sqrt{N}}\sum_{n=0}^{N-1} \mathbf{x}_i^r(n)e^{-\omega_k n}$ is computed. Let $C = \{c_1, c_2, \ldots, c_m\} \subseteq V$ where $V$ is the set of nodes. Let $\mathbf{X}_C^r(\omega_k) \in \mathbb{C}^m$ be the vector $\begin{bmatrix} \mathbf{X}_{c_1}^r(\omega_k) & \mathbf{X}_{c_2}^r(\omega_k) \ldots & \mathbf{X}_{c_m}^r(\omega_k) \end{bmatrix}^\top$ obtained by stacking the Fourier coefficients from the $r^{th}$ segment of the time-series in the set $C$. Let $\mathcal{X}_i(\omega_k) := [\mathbf{X}_i^1(\omega_k)\ \ldots\ \mathbf{X}_i^R(\omega_k)]^\top \in \mathbb{C}^R$ and let $\mathcal{X}_C(\omega_k) := [\mathbf{X}_C^1(\omega_k)\ \ldots\ \mathbf{X}_C^R(\omega_k)]^\top \in \mathbb{C}^{R \times m}$, where $C \subseteq V$. Then, for any $\omega_k \in \Omega_N$, the least square formulation described by,

$$\mathbf{W}_{i \cdot C}^{(f)}(\omega_k) := \arg \min_{\beta \in \mathbb{C}^{|C|}} \frac{N}{T}\|\mathcal{X}_i(\omega_k) - \mathcal{X}_C(\omega_k)\beta\|_2^2, \tag{6}$$

computes an estimate of the WF coefficients in the FD (see Doddi et al. (2022) for details). Notice that Doddi et al. (2022) provided the FD computation of the Lasso-based representation of the WF for a special case of our general model. In this article we emphasize on 1) the computational advantages of employing FFT, and 2) the non-penalized MMSE estimate of the WF. In Appendix D.1 an explicit convergence result of FFT to DTFT is provided, which shows a convergence rate of $1/\sqrt{N}$. The sample complexity of estimating $W_{i \cdot C}^{(f)}(\omega_k)$ scales as $O(Nn)$ as shown in Theorem 5.2.

## 2.6 Computational Complexity of Estimating WF

The first major contribution of the article is the characterization of the computational efficiency in computing the multivariate WFs in FD when compared to the TD projections. In the TD, for $m = |C|$,

the dimension of $\mathbf{y}_c$ in equation 4 is $(T - 2L + 1) \times (2L + 1)m$. Here the complexity of computing the least square estimate for node $i$ using equation 4 is $O((T - 2L + 1)m^2(2L + 1)^2) \approx O(Tm^2L^2)$, since $L < T$. On the other hand, computing the FFT for a window size of $N$ samples takes $N \log N$ computations Oppenheim & Verghese (2017). Notice that we compute the FFT with $N$ samples per segment, and so the number of effective samples in the regression computation of equation 6 is $R = T/N$. Thus the dimension of $\mathcal{X}_C(\omega_k)$ is $(T/N) \times m$. Computing the Wiener co-efficient for a given $\omega_k$ using the regression equation 6 takes $O(\frac{Tm^2}{N})$ computations, thus a total of $O(Tm^2 \log N)$ computations are required to compute the Wiener coefficient for a single frequency. It follows that the FFT-based computation provides $O(L^2 / \log N)$ improvement compared to TD computation in the asymptotic worst case computation complexity. For the settings with large number of samples (large $T$) or with longer delays (large $L$) the improvement is significant.

## 3 IDENTIFICATION OF MARKOV EQUIVALENCE GRAPH OF SPS

In the previous preliminaries section we described two methods of determining the WF coefficients from finite data; the first via projections in TD and the second via projections in the FD. We would like to emphasize that the focus of the article is on detection rather than estimation wherein we are interested in determining whether a WF coefficient is zero or not. The WF coefficient being zero or not will be used to retrieve the causal structure corresponding to the underlying data generative process. In general, it is not possible to reconstruct the exact and the complete structure of the directed graph $G$, (the generative graph), associated with an LDIM from data alone without actively intervening. Instead, in many scenarios the best one can do is to retrieve the Markov Equivalence Graph (MEG) which is the set of graphs that satisfy the same conditional dependence property as that of the true generative DAG Ghoshal & Honorio (2018).

### 3.1 REALIZING THE ESSENTIAL GRAPH OF SPS USING WIENER FILTERING

In the static DAGs, reconstruction of MEG is well explored Spirtes et al. (2000). One of the popular approaches in the static setting is the Peter-Clarke (PC) algorithm, which is performed using pairwise conditional independence (CI) tests Kalisch & Bühlman (2007). In static settings, tests such as the Fischer-z test and the chi-square test are performed to assess CI. However, the static CI tests fail in the dynamic setting because of the temporal dependency in the time-series as shown in Table 1 in the experimental results section. However, WF-based techniques can be applied on time-series with dynamical dependencies to obtain better results, as suggested by the following result from Materassi & Salapaka (2013).

**Lemma 3.1** (Materassi & Salapaka (2013)). *Consider a well-posed LDIM given by equation 2. Let $i, j \in V$ and let $Z \subseteq V \setminus \{i, j\}$. Let $W_{i \cdot [j, Z]}$ be the WF coefficients of estimating the time-series $x_i$ from the time-series $x_j$ and $x_Z$. If $i$ and $j$ are d-separated given $Z$ in $G = (V, \vec{\mathcal{E}})$, then $W_{i \cdot [j, Z]}[j](\omega) = 0$, for every $\omega \in \Omega$. The converse holds almost always.*

Inspired by the above result, we implement a modified PC algorithm, where the WF is employed to test the CI on data with dynamical dependencies, to reconstruct the MEG. This proposed algorithm, called Wiener-PC (W-PC) is provided in Algorithm 1, where the CI is performed by applying Lemma 3.1. Recall that the WF can be computed using either equation 4 or equation 6.

The following are the notations used in Algorithm 1. $q$ denotes the maximum d-separating set size. $combinations(D, k)$ lists all the $k$-length tuples in $D$, in a sorted order, without any repeated elements present, and $DS(i, j)$ denotes the d-separating set identified for $i$ and $j$.

### 3.2 COMPUTATIONAL COMPLEXITY OF W-PC

As shown in Kalisch & Bühlman (2007), PC algorithm requires $O(n^q)$ tests. From Section 2.6, we have that computing $W_{i \cdot C}^{(f)}(\omega)$ using FFT takes a complexity of $O(T|C|^2 \log N) \approx O(Tn^2 \log N)$, since $|C| < n$. Thus the overall worst case complexity of W-PC when implemented in FD using equation 6 is $O(Tn^{q+3} \log N)$, in contrast to the TD approach, which has the complexity $O(Tn^{q+3}L^2)$.

## 4 WIENER-PHASE ALGORITHM

In this section, we propose a computationally attractive algorithm to reconstruct the MEG by exploiting certain properties of the WF, which are not present in any prior static approaches. Towards

this, we provide some useful lemmas and preliminaries below. The following Lemma shows that the set of kins in $G$ can be reconstructed from the support of WF.

**Lemma 4.1** (Materassi & Salapaka (2012)). *Consider a well-posed LDIM given by equation 2. Let $W_{i \cdot \bar{i}}$, where $\bar{i} = V \setminus \{i\}$, be the WF coefficients of estimating the time-series $x_i$ from the time-series $x_{\bar{i}}$. If $W_{i \cdot \bar{i}}[j](\omega) \neq 0$ then $i \in kin_G(j)$. The converse holds almost always.*

In many applications (see Remark 17 in Veedu et al. (2021)), it is possible to retrieve $Skel(G)$ exactly from the imaginary part of the WF. The following assumptions are required to obtain this result.

*Assumption* 1. If a node $k$ has multiple incoming edges in $G$, then for every pair of in-nodes $i, j$ of $k$ and for every $\omega \in \Omega$, $\angle \mathbf{H}_{ki}(\omega) = \angle \mathbf{H}_{kj}(\omega)$, where $i, j, k \in V$.

*Assumption* 2. If $\mathbf{H}_{ij}(\omega) \neq 0$ then $\Im\{\mathbf{H}_{ij}(\omega)\} \neq 0$, for every $i, j \in V$ and $\omega \in \Omega$.

Assumption 1 says that the phase angle of all the incoming edges must be the same. This assumption seems restrictive; surprisingly a large class of problems satisfy this assumption. Applications satisfying Assumption 1 are provided in the supplementary material. Assumption 2 is required for the consistency of the reconstruction of the skeleton.

**Lemma 4.2.** *Veedu et al. (2021) Consider a well-posed LDIM given by equation 2 and satisfying Assumptions 1 and 2. For any $\omega \in \Omega$, if $\Im\{W_{i \cdot \bar{i}}[j](\omega)\} \neq 0$ then either $(i, j) \in \vec{\mathcal{E}}$ or $(j, i) \in \vec{\mathcal{E}}$. The converse holds almost always.*

Thus, by analyzing the real and imaginary part of $W_{i \cdot \bar{i}}$ separately, one can extract the strict spouse edges. This information can in turn be employed to identify the colliders in $G$ in an efficient way. We call this algorithm (Algorithm 2) Wiener-Phase here. The worst case asymptotic computational complexity of Wiener-Phase algorithm is $O(n^3(T \log N + q^2))$ (see Section 4.1), which is advantageous, especially in highly connected graphs (with large $q$). We demonstrate the advantages of the Wiener-phase algorithm in Section 6.

## 4.1 ANALYSIS OF THE WIENER-PHASE ALGORITHM

In here, we analyze the Wiener-Phase algorithm (Algorithm 2). As shown in Lemma 4.2, in many applications Veedu et al. (2021), support of the imaginary part of the frequency dependent WF retrieves $Skel(G)$ exactly. Further, as shown in Lemma 4.1, the support of $W_{i \cdot \bar{i}}$ retrieves the Markov blanket structure in $G$. Combining both, the Wiener-Phase algorithm identifies the colliders and thus the MEGs efficiently. Here, $\mathcal{K}$ is the set of kin edges obtained using Lemma 4.1 and $\mathcal{S}$ is the skeleton obtained from Lemma 4.2. Consider any edge $(i, j)$ that belongs to $\mathcal{K}$ but does not belong to $\mathcal{S}$. Then $i$ and $j$ share a common child without a direct link between them (thus are strict spouses) in $G$.

---

**Algorithm 1** Wiener-PC algorithm

**Input:** Data $\mathcal{X}(\omega), \omega_k \in \Omega_N, q$
**Output:** $\widehat{G}$

1. Initialize the sets, $\mathcal{S} \leftarrow$ fully connected undirected edge set, $Col \leftarrow \{\}, DS \leftarrow \{\}, E_{est} \leftarrow ()$

2. For $i = 1, \dots, n$

   (a) For $j = 1, \dots, n$, and $j < i$

      i. Initialize $D \leftarrow V \setminus \{i, j\}$

      ii. For $k = 0, \dots, q - 1$

         • For $z$ in $combinations(D, k)$

           – If $\left| \mathbf{W}_{i \cdot [j,z]}^{(f)}[j](\omega_k) \right| < \tau$

              ∗ $DS(i, j) \leftarrow z$

              ∗ delete $(i, j)$ from $\mathcal{S}$

              ∗ **break**

           – If $|z| >= q - 1$

              ∗ $DS(i, j) \leftarrow -1$

3. For $(i, j) \in V \times V$ and $i < j$:

   • If $\{i, j\} \in \mathcal{S}$

      – $E_{est}(i, j) \leftarrow 1$

      – $E_{est}(j, i) \leftarrow 1$

4. For $(i, j) \in V \times V$ and $i < j$

   (a) For $k = j + 1, \dots, n$

      i. If $\{i, k\} \in \mathcal{S}$ & $\{i, j\} \in \mathcal{S}$ & $i \notin \{j, k\}$

         • If $DS(j, k) > -1$ & $i \notin DS(j, k)$

           – $Col \leftarrow Col \cup \{i\}$

           – $E_{est}[j, i] \leftarrow 0$

           – $E_{est}[k, i] \leftarrow 0$

5. For every undirected edges, i.e., if $(i, j) \in E_{est}$ and $(j, i) \in E_{est}$, fix the orientation of as many as possible such that 1) there are no directed cycles 2) there are no collider.

6. Return $\widehat{G} = (V, E_{est})$

---

Following this procedure, we can construct all the strict

spouses. Let this set be $\mathcal{SP}$. Recall from Section 2.6 that the complexity of computing $W_{i\cdot\bar{i}}^{(f)}(\omega)$ is $O(n^2 T \log N)$. Repeating this process $n$ times gives $O(n^3 T \log N)$. Thus, the total complexity in finding $\mathcal{K}$ and $\mathcal{S}$ using FFT approach is $O(n^3 T \log N)$.

Now consider the skeleton $\mathcal{S}$. In Step 4, we identify the common child between the strict spouses $i, j$ as follows. For any $\{i, j\} \in \mathcal{SP}$, let $\mathcal{C}_{ij} := \{k : \{i, k\} \in \mathcal{S}, \{j, k\} \in \mathcal{S}\}$, which is a set that contains all nodes $k$ which has a link to both $i$ and $j$ in the skeleton. Note that the set $\mathcal{C}_{ij}$ can contain non-colliders, which will be eliminated using the following steps. For any $\{i, j\} \in \mathcal{SP}$, if $|\mathcal{C}_{ij}| = 1$, then $c \in \mathcal{C}_{ij}$ is a collider (because the link between $i$ and $j$ in $\mathcal{K}$ is formed by at least one collider). If $|\mathcal{C}_{ij}| > 1$, then for every $c \in \mathcal{C}_{ij}$, we can compute $W_{i\cdot[j,c]}^{(f)}[j]$. If $W_{i\cdot[j,c]}^{(f)}[j] \neq 0$, then $c$ is a collider since $\{i, j\} \notin \mathcal{S}$. The complexity of computing $\mathcal{K}$ and $\mathcal{S}$ is $O(n^3 T \log N)$. Step 4 is repeated $O(nq)$ times and Step 4(a) which checks for potential colliders among the common neighbors of $i$ and $j$ takes $O(n^2 q)$ operations. Steps 4(b) and 4(c) take $O(n)$. Thus the complexity in computing Step 4 is $O(n^3 q^2)$,

---

**Algorithm 2** Wiener-Phase algorithm

**Input:** Data $\mathcal{X}(\omega), \omega_k \in \Omega_N$
**Output:** $\widehat{G}$

1. Initialize the ordering, $\mathcal{S} \leftarrow ()$
2. For $i = 1, \ldots, n$
   (a) Compute $W_{i\cdot\bar{i}}^{(f)}(\omega)$ using equation 6
   (b) For $j = 1, \ldots, n$
       i. If $|W_{i\cdot\bar{i}}^{(f)}[j](\omega_k)| > \tau$, then $\mathcal{K} \leftarrow \mathcal{K} \cup \{i, j\}$
       ii. If $|\Im\{W_{i\cdot\bar{i}}^{(f)}[j](\omega_k)\}| > \tau$, then $\mathcal{S} \leftarrow \mathcal{S} \cup \{i, j\}$
3. Compute $\mathcal{SP} := \mathcal{K} \setminus \mathcal{S}$
4. For $\{i, j\} \in \mathcal{SP}$
   (a) For $k = 1, \ldots, n$
       i. If $\{i, k\} \in \mathcal{S}$ and $\{j, k\} \in \mathcal{S}$
          • $C_{ij} \leftarrow C_{ij} \cup \{k\}$
   (b) If $|C_{ij}| = 1$: $Col \leftarrow Col \cup \{k\}$
   (c) Else: for $c \in C_{ij}$ compute $W_{i\cdot[j,c]}^{(f)}$
       i. If $|W_{i\cdot[j,c]}^{(f)}[j]| > \tau$: $Col \leftarrow Col \cup \{c\}$
5. Return $\widehat{G}$

---

giving the overall complexity of $O(n^3(T \log N + q^2))$. It is sufficient to compute Algorithm 2 for $O(1)$ number of $\omega$. Thus, the final complexity remains same.

## 5 SAMPLE COMPLEXITY ANALYSIS

So far, we have discussed about estimation of WFs using TD and FD approaches and its use in causal inference algorithms. However, it is to be noted that in practice, due to finite sample effects, the estimated WFs using either equation 4 or equation 6 will deviate from the true WF shown in equation 3. A quantification of these errors is necessary for better understanding and implementation of WF-based algorithms. In this section, we characterize the error in estimating the WF coefficients from data, and consequently, the number of samples required to estimate the WF reliably. The first step in bounding the error in the WF estimation is bounding the error in estimating the PSDM.

The error in estimating the PSDM in TD is a special case of the Blackman-Tukey estimator (Veedu et al. (2021), Lamperski (2023)). They compute the Fourier transform of the autocorrelation estimates to obtain the PSDM estimates. The following theorem provides sample complexity for estimating $W_{i\cdot C}$ from equation 4, by using the PSDM concentration bound from Lamperski (2023). The proof is provided in Appendix B.

**Theorem 5.1.** *Consider a linear dynamical system governed by equation 2. Suppose that the autocorrelation function $R_\mathbf{x}(k)$ satisfies exponential decay, $\|R_x(k)\| \leq C\rho^{-|k|}$ and that there exists $M$ such that $\frac{1}{M} \leq \lambda_{min}(\Phi_\mathbf{X}) \leq \lambda_{max}(\Phi_\mathbf{X}) \leq M$. Then for any $0 < \epsilon, \delta$ and $L \geq \log_\rho\left(\frac{(1-\rho)\epsilon}{2C}\right)$ and $\omega \in \Omega$, if*

$$T \gtrsim O\left(\frac{(6n - \log(\delta))\left((2L+1)M^4\right)}{\epsilon^2} + L\right), \text{ then } \forall \omega \in \Omega, \mathbb{P}\left(\|\mathbf{W}_{i\cdot C} - \widehat{\mathbf{W}}_{i\cdot C}^{(T,L)}\| > \epsilon\right) < \delta.$$

The following theorem, based on the PSDM estimator in Veedu et al. (2024) provides the sample complexity for estimating $\mathbf{W}_{i\cdot C}$ from equation 6. The proof is provided in the Appendix B.

**Theorem 5.2.** *Consider a linear dynamical system governed by equation 2. Suppose that the autocorrelation function $R_x(k)$ satisfies exponential decay, $\|R_x(k)\| \leq C\rho^{-|k|}$ and that there exists*

*M such that $\frac{1}{M} \leq \lambda_{min}(\Phi_{\mathbf{X}}) \leq \lambda_{max}(\Phi_{\mathbf{X}}) \leq M$. Let $0 < \varepsilon_1, \varepsilon_2 < \varepsilon$ be such that $\varepsilon_2 = \varepsilon - \varepsilon_1$. Suppose that $N > \frac{2C\rho^{-1}}{(1-\rho^{-1})^2\varepsilon_1}$. Then $\forall \omega \in \Omega$ and $0 < \delta$, if*

$$T \gtrsim O\left(\frac{(6n - \log(\delta))\,NM^4}{\varepsilon^2}\right), \text{ then } \forall \omega \in \Omega, \ \mathbb{P}\left(\|\mathbf{W}_{i \cdot C}(\omega) - \widehat{\mathbf{W}}_{i \cdot C}^{(f)}(\omega)\| > \varepsilon\right) < \delta.$$

Comparing Theorem 5.1 with Theorem 5.2, it can be observed that number of samples required for a given error performance is comparable when $N \approx L$. That is, $T = O(Ln)$ for the TD whereas $T = O(Nn)$ for the FD computation. In addition, simulation results in Section 6 show that the FD approach shows better reconstruction accuracy than TD approach at smaller range of samples.

## 6 EXPERIMENTAL RESULTS

In this section, we demonstrate that FFT-based approach provides superior computational advantages while having comparable reconstruction accuracy, if not better, in networks with dynamics using synthetic data as well as real field data.

The data generation and algorithm execution were performed in a computer with intel core i9-14900K, 3200 Mhz, 24 cores, 32 logical processors. The generative model for the synthetic datasets were vector auto-regressive (VAR) model as shown below

$$x_i(t) + a_i(1)x_i(t-1) + a_i(2)x_i(t-2) + a_i(3)x_i(t-3) = \sum_{j \neq i} b_{ij}x_j(t-1) + e_i(t), \quad (7)$$

where $i \in \{1, 2, ..., 6\}$, and $e_i(\cdot)$ are zero mean i.i.d Gaussian noise with diagonal PSDM and $e_i(t) \sim N(0,1)$. The coefficients of the VAR model were generated in accordance with the graphical structure of the corresponding DAGs i.e. $b_{ij}$ was set to zero if a link from $j$ to $i$ does not exist in the corresponding DAG. The non-zero $b_{ij}$ coefficients in the VAR model were randomly generated with $b_{ij} \sim U(0.2, 0.4)$, and $a_i(\cdot)$ were fixed. A more detailed description of the data generation process is given in

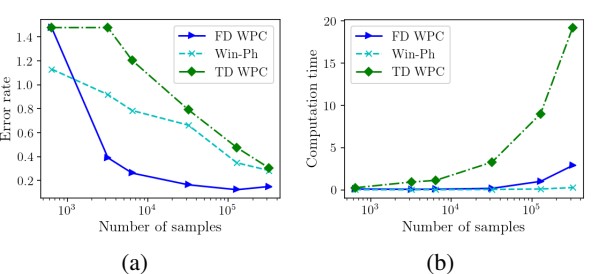

(a)        (b)

Figure 1: Comparison of (a) average error rate (b) average computation time for 25 random networks generated according to equation 7

Appendix C. For the FD based algorithms we averaged the magnitude and phase of the Wiener filters, obtained from data, over a set of frequency $\Psi = \{\omega_1, ...\omega_\alpha\}$, where $N \geq \alpha \in \mathbb{N}$. The average magnitudes were then compared to the chosen thresholds in the algorithms.

### 6.1 RANDOM NETWORKS

Fig 1(a) shows the average error rate for 25 randomly generated DAGs with dynamics described by equation 7 for the three algorithms, namely, W-PC with TD-based WF estimation, W-PC with FD-based WF estimation, and Wiener-phase algorithm. Here, error rate is the ratio of the total number of wrong edges (both false positive and false negative) to the number of edges in the actual DAG. It can be observed that FD-based W-PC outperforms the TD-based W-PC in terms of average error. Moreover, the FD-based W-PC algorithm requires much smaller sample size compared to the TD-based W-PC algorithm for similar reconstruction accuracy.We note that the Wiener-phase algorithm also outperforms the TD-based W-PC algorithm. It can be observed that the average computation time for the TD-based W-PC algorithm is significantly larger compared to the FD-based W-PC and Wiener-phase algorithms (see Fig. 1(b)). This difference is stark at higher number of samples where there is around an order of magnitude difference between the computation time required for the TD-based W-PC and the FD based W-PC.

To choose the thresholds for the randomly generated network we use a sparsity metric based approach. Bounds on sparsity metrics such as average, maximum, and minimum degree are typically available apriori. As described in Appendix C.2, given a dataset and an algorithm, we use an iterative search method to obtain the smallest value of thresholds that produce networks which satisfies the given bounds on average degree, and minimum degree.

Table 1: Comparison of algorithms on 20 nodes network and river-runoff dataset

| Algorithm | 20 Nodes synthetic dataset | | | | | Real-world river-runoff dataset | | | | |
|---|---|---|---|---|---|---|---|---|---|---|
| | $F_1$ | $CS$ | $TPR$ | $\widetilde{FPR}$ | Run Time | $F_1$ | $CS$ | $TPR$ | $\widetilde{FPR}$ | Run Time |
| Fisher-Z PC | 24.44 | 21.73 | 35.48 | 86.25 | 0.625 | 52.63 | 64.32 | 76.92 | 87.4 | 0.526 |
| CD-NOD | 23.80 | 19.93 | 32.25 | 87.68 | 0.421 | 54.05 | 65.16 | 76.92 | 88.24 | 0.1794 |
| GC | 48.65 | 50.9 | 58.06 | 92.84 | 0.939 | 33 | 28.37 | 38.46 | 89.92 | 0.22 |
| TPC | 78.57 | 70.12 | 70.97 | 99.15 | 4016.57 | 43.64 | 67.1 | 92.31 | 74.79 | 1548.29 |
| FFT-WPC | 100 | 100 | 100 | 100 | 1011.62 | 75 | 67.5 | 69.2 | 98.3 | 9.041 |
| Wiener Phase | 93.55 | 92.97 | 93.54 | 99.43 | 1.37 | 48.3 | 46.29 | 53.85 | 92.44 | 0.02593 |

## 6.2 COMPARISON WITH STATE OF THE ART METHODS AND SCALABILITY

To further demonstrate the efficacy we present comparative results for network with 20 nodes. The details of the network under consideration is given in Appendix C.3. We compare the performance of our FD method based algorithms with state of the art methods, namely, Fisher-Z test based PC Kalisch & Bühlman (2007), CD-NOD Huang et al. (2020), Granger Causality (GC) Granger (1969), and Time Aware PC (TPC) Biswas & Shlizerman (2022), on synthetic data. The Fisher-Z PC, and CD-NOD algorithms were obtained from causal-learn library Zheng et al. (2024), and the TPC and GC algorithms were obtained from the TimeAwarePC library Biswas & Shlizerman (2022). To compare the algorithms, we define the following metrics: True positive rate ($TPR$) $=\frac{TP}{TP+FN} \times 100\%$, 1$-$False Positive Rate ($\widetilde{FPR}$) $= \left(1 - \frac{FP}{FP+TN}\right) \times 100\%$, Combined Score ($CS$) $= TPR - FPR$, and F-1 score ($F_1$) $= \frac{2TP}{2TP+FP+FN} \times 100\%$, where $TP$, $FP$, $TN$, and $FN$ stand for $True\ Positive$, $False\ Positive$, $True\ Negative$, and $False\ Negative$ respectively. We also report the run times in seconds. A dataset containing 128000 samples was generated using the model of equation 7 for a 20 nodes network with 22 edges. The accuracy metrics and the runtime of each algorithm using the dataset of 20 nodes is shown in Table-1. The thresholds for the FFT-WPC and Wiener Phase algorithms were chosen to be 0.034 in both algorithms which were determined using the sparsity based threshold tuning algorithm in Appendix C.2. For the GC and Fisher-Z PC significance levels of 0.106 and 0.01 were used respectively. The TPC algorithm was implemented with maximum time-delay of interaction to be 5, and a significance level $\alpha = 0.1$ for kernel-based conditional independence tests. For the bootstrap procedure in TPC, 50 bootstrap iterations with bootstrap window length of 50 recordings and bootstrap stability threshold $\gamma = 0.01$ were chosen. It can be observed that FFT-WPC and Wiener Phase algorithms outperform TPC, Fisher-Z based PC, and GC in terms of accuracy. Also observe that FFT based WPC beats TPC in terms of speed also. Moreover Wiener-phase beats both TPC and GC in speed of execution.

To further examine the scalability of the proposed algorithms, we applied the FFT based Wiener-PC and the Wiener phase algorithm to a network with 50 nodes and 54 edges. The details of the generative graph is given in Appendix C.4. The FFT based Wiener PC algorithm with a threshold of 0.035 resulted in $TPR = 97.62\%$, $(1 - FPR) = 99.88\%$, and $CS = 97.5\%$, with a total runtime of 142652 seconds ($\approx 40$ hours). The Wiener Phase algorithm with a threshold of 0.051 resulted in $TPR = 100\%$, $(1 - FPR) = 99.83\%$, and $CS = 99.83\%$, with a total runtime of 19.49 seconds.

## 6.3 RESULTS ON REAL FIELD DATA

In this section we demonstrate the performance of the proposed method on a real-world benchmark dataset, the river-runoff dataset, obtained from online repository of CauseMe Runge et al. (2019). The river runoff data has 4600 samples of average daily river runoff of the upper Danube basin Ji et al. (2024). The graphical interpretation of the river runoff data was taken from Ji et al. (2024) as the ground truth. The dataset was used in the FFT-WPC, Wiener Phase, Fisher-Z test based PC, CD-NOD, GC, and TPC algorithms. For the FFT-WPC and Wiener Phase the thresholds were chosen to be 0.404 and 0.108. The significance level parameter for GC and Fisher-Z-PC were taken to be 0.112 and 0.01 respectively. The parameters for the TPC algorithm were taken from Appendix-D of Biswas & Shlizerman (2022). The comparative results are shown in Table 1. Observe that, FFT-WPC was able to obtain comparable error performance with the best algorithm reported in Biswas & Shlizerman (2022), TPC, while run time was orders of magnitude faster. Wiener phase was the fastest with a runtime of 26 milliseconds, but the CS was lower than FFT-WPC and TPC. However, Wiener phase was able to beat GC in CS and F-1 score.

## 6.4 APPLICATION ON CIRCUITS

In this section, we demonstrate the application of the proposed methods in circuits that employ transistors, capacitors, and resistors. In Rana et al. (2025) it has been shown that electronic circuits can be represented as LDIM. To demonstrate the effectiveness of the proposed algorithms in this

Table 2: Comparison of algorithms on analog electronic circuit dataset

| Algorithm | MOSFET Hardware Circuit Dataset | | | | | BJT Hardware Circuit Dataset | | | | |
|---|---|---|---|---|---|---|---|---|---|---|
| | $F_1$ | $CS$ | $TPR$ | $\widetilde{FPR}$ | Run Time | $F_1$ | $CS$ | $TPR$ | $\widetilde{FPR}$ | Run Time |
| Fisher-Z PC | 40 | 11.11 | 66.67 | 44.44 | 0.021 | 40 | 0 | 100 | 0 | 0.019 |
| CD-NOD | 66.67 | 66.67 | 100 | 66.67 | 0.0323 | 40 | 0 | 100 | 0 | 0.0309 |
| GC | 85.71 | 88.89 | 100 | 88.89 | 0.0389 | 75 | 77.78 | 100 | 77.78 | 0.0391 |
| TPC | 75 | 77.78 | 100 | 77.78 | 20.816 | 60 | 55.56 | 100 | 55.56 | 23.641 |
| FFT-WPC | 100 | 100 | 100 | 100 | 0.0842 | 100 | 100 | 100 | 100 | 0.0798 |
| Wiener Phase | 100 | 100 | 100 | 100 | 0.0464 | 100 | 100 | 100 | 100 | 0.0419 |

article over state-of-the-art methods, we collected two datasets, one containing measured data from bipolar junction transistor (BJT) based physical hardware set-up and another containing measured data from metal-oxide-semiconductor field-effect transistor (MOSFET) based hardware set-up.

First, we present results on the dataset collected from BJT based physical hardware circuit implemented on a printed circuit board (PCB). Fig. 2 shows the circuit schematic along with the PCB and generative graph. To develop the circuit, MMBT2222ATT1G BJT units were used along with resistors and capacitors. The node voltages $\{v_1,...v_4\}$ were measured at 1MS/s rate using a Ni-cRio data acquisition system and AD8421 instrumentation amplifiers. A detailed description of the hardware set-up is given in Appendix C.9. A total of 500000 samples were collected for each node voltage. The results of reconstruction are shown in Table 2. Observe that the proposed method outperforms state-of-the-art methods in terms of both accuracy and speed.

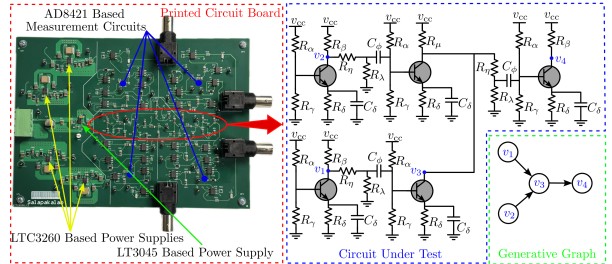

Figure 2: Schematic of BJT based hardware implemented circuit showing PCB implementation, circuit under test, and generative graph

Next, we present results on the dataset collected from a MOSFET based hardware circuit implemented on a printed circuit board (PCB). Fig. 3 shows the circuit schematic along with the PCB and generative graph. To develop the circuit, 2N7002E n-channel MOSFETS units from Onsemi were used along with resistors and capacitors. The node voltages $\{v_1,...v_4\}$ were measured at 1MS/s rate using the set up described in the previous section. A total of 500000 samples were collected for each node voltage. The results of reconstruction are shown in Table 2. Observe that similar to the BJT circuit results, the proposed method outperforms state-of-the-art methods in terms of both accuracy and speed. This demonstrates the dominance of our proposed method for the tested real-world dynamical systems.

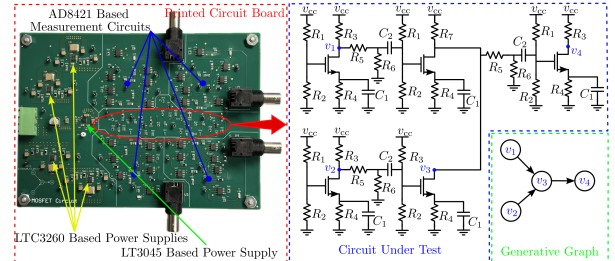

Figure 3: Schematic of MOSFET based hardware implemented circuit showing PCB implementation, circuit under test, and generative graph

## CONCLUSIONS

This article answers the question of whether time-domain (TD) or frequency-domain (FD) approach to reconstruct causal structure of dynamical networks is better. It is established through theory and simulation results that FD approach is better than the TD approach in terms of computational complexity. Further, simulation results show that the FD approach outperforms TD. In addition, the superiority of the WF-based causal inference over traditional static CI test-such as Fischer-Z test, and state of the art methods such as Time aware PC, and Granger Causality is demonstrated through simulation. As a third contribution, an algorithm for causal structure recovery using magnitude and phase properties of the WFs is developed. This algorithm provides significant computational advantages over traditional causal inference algorithms by ruling out the combinatorial explosion of required CI tests.

ACKNOWLEDGMENTS

This work was supported by National Science Foundation via grant number ECCS 2412435 and by United States Department of Energy via grant number DE-CR0000040.

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

APPENDIX

## A WHICH FREQUENCY TO CHOOSE?

Most practical systems are finite dimensional with finite dimensional realizations. These admit rational transfer functions. The following Lemma proves that, in identifying the CI, graph/moral graph, and skeleton, it is sufficient to work with one frequency selected arbitrarily, in order to get meaningful result, when we have access to large enough data.

**Lemma A.1.** *Consider any rational polynomial transfer function*

$$\mathbf{H}_{ij}(z) = C \frac{(z - a_1)(z - a_2)\dots(z - a_p)}{(z - b_1)(z - b_2)\dots(z - b_q)}, \; p \le q, \; C \in \mathbb{C}.$$

*For any distinct $\omega_1, \omega_2 \in \Omega$, $\mathbf{H}_{ij}(e^{j\omega_1}) \ne 0$ if and only if $\mathbf{H}_{ij}(e^{j\omega_2}) \ne 0$ almost surely w.r.t. a continuous probability measure.*

**Proof:** By the fundamental theorem of algebra Cox et al. (2007), there exists at most $p$ complex zeros for $\mathbf{H}_{ij}(z)$, if $\mathbf{H}_{ij}(z)$ is not identically zero, which form a set of Lebesgue measure zero. Suppose $\mathbf{H}_{ij}(e^{j\omega_1}) \ne 0$, which implies that $\mathbf{H}_{ij}$ is not identically zero. Thus, $\mathbf{H}_{ij}(e^{j\omega_2}) \ne 0$ almost everywhere. ∎

## B PROOF OF THE RESULTS IN SECTION 5

$$\begin{aligned}
\|\mathbf{W}_{i\cdot C} - \widehat{\mathbf{W}}_{i\cdot C}\| &= \|\widehat{\Phi}_{iC}\widehat{\Phi}_{CC}^{-1} - \Phi_{iC}\Phi_{CC}^{-1}\|_2 \\
&\le \|\widehat{\Phi}_{iC}(\widehat{\Phi}_{CC}^{-1} - \Phi_{CC}^{-1})\|_2 + \|(\widehat{\Phi}_{iC} - \Phi_{iC})\Phi_{CC}^{-1})\|_2 \\
&\le \|\widehat{\Phi}_{iC}\|_2\|\widehat{\Phi}_{CC}^{-1} - \Phi_{CC}^{-1}\| + \|\widehat{\Phi}_{iC} - \Phi_{iC}\|_2\|\Phi_{CC}^{-1}\| \\
&\le \|\widehat{\Phi}_{iC}\|_2\|\widehat{\Phi}_{CC}^{-1} - \Phi_{CC}^{-1}\| + M\|\widehat{\Phi}_{iC} - \Phi_{iC}\|_2 \\
&\le (\|\widehat{\Phi}_{iC} - \Phi_{iC}\|_2 + \|\Phi_{iC}\|_2)\|\widehat{\Phi}_{CC}^{-1} - \Phi_{CC}^{-1}\| + M\|\widehat{\Phi}_{iC} - \Phi_{iC}\|_2 \\
&= \|\widehat{\Phi}_{iC} - \Phi_{iC}\|_2\|\widehat{\Phi}_{CC}^{-1} - \Phi_{CC}^{-1}\| + \|\Phi_{iC}\|_2\|\widehat{\Phi}_{CC}^{-1} - \Phi_{CC}^{-1}\| \\
&\quad + M\|\widehat{\Phi}_{iC} - \Phi_{iC}\|_2 \\
&\le \|\widehat{\Phi}_{CC} - \Phi_{CC}\|\|\widehat{\Phi}_{CC}^{-1} - \Phi_{CC}^{-1}\| + M\|\widehat{\Phi}_{CC}^{-1} - \Phi_{CC}^{-1}\| \\
&\le \epsilon_1.\epsilon_{inv} + M\epsilon_{inv} \le (M+1)\epsilon,
\end{aligned}$$

where we have used triangle inequality, properties of spectral norm, $\|\cdot\|$ and Euclidean norm $\|\cdot\|_2$.

From Section 5.8 in Horn & Johnson (2012),

$$\|\Phi_{CC} - \widehat{\Phi}_{CC}\| \le \|\Phi_{CC}\|\|\Phi_{CC}^{-1}\|^{-1}\|\widehat{\Phi}_{CC}^{-1} - \Phi_{CC}^{-1}\| \frac{M^2}{1 - M^2 \frac{\|\widehat{\Phi}_{CC}^{-1} - \Phi_{CC}^{-1}\|}{\|\Phi_{CC}^{-1}\|}}$$

$$\le \frac{M^4\|\widehat{\Phi}_{CC}^{-1} - \Phi_{CC}^{-1}\|}{1 - M\|\widehat{\Phi}_{CC}^{-1} - \Phi_{CC}^{-1}\|} \le \epsilon \implies \|\widehat{\Phi}_{CC}^{-1} - \Phi_{CC}^{-1}\| \le \frac{\epsilon}{M^4 + M\epsilon} \le \frac{\epsilon}{M^4}.$$

Therefore, to guarantee that $\|\widehat{\Phi}_{CC}^{-1} - \Phi_{CC}^{-1}\| < \epsilon$, it is sufficient to guarantee that $\|\widehat{\Phi}_{CC} - \Phi_{CC}\| < \epsilon$ since $M \ge 1$. Hence,

$$\mathbb{P}\left(\|\mathbf{W}_{i\cdot C} - \widehat{\mathbf{W}}_{i\cdot C}\| > \epsilon\right) \le \mathbb{P}\left(\|\Phi_{CC} - \widehat{\Phi}_{CC}\| > \frac{\epsilon}{M+1}\right). \tag{8}$$

For Blackman-Tuckey,

**Lemma B.1** (Lamperski (2023)). *Consider a linear dynamical system governed by equation 2. Suppose that the auto-correlation function $R_x(k) := \mathbb{E}\{X(n)X^\top(n+k)\}$ satisfies exponential decay, $\|R_x(k)\| \le C\rho^{-|k|}$ and that there exists $M$ such that $\frac{1}{M} \le \lambda_{min}(\Phi_{\mathbf{X}}) \le \lambda_{max}(\Phi_{\mathbf{X}}) \le M$. Then for any $0 < \epsilon$ and $L \ge \log_\rho\left(\frac{(1-\rho)\epsilon}{2C}\right)$,*

$$\mathbb{P}\left(\|\widehat{\Phi}_{CC}^{(T,L)} - \Phi_{CC}\| > \epsilon\right) \le \exp\left(-\frac{\epsilon^2(T-L)}{32(2L+1)M^2} + 6n\right)$$

Table 3: Coefficients used in data generation for random networks experiment of Section 6.1

| | $k = 1$ | $k = 2$ | $k = 3$ |
|---|---|---|---|
| $a_1(k)$ | $1.739381261594696859 \times 10^{-2}$ | $1.043628756956818150 \times 10^{-2}$ | $5.218143784784090751 \times 10^{-3}$ |
| $a_2(k)$ | $3.415208172002098808 \times 10^{-1}$ | $1.366083268800839523 \times 10^{-1}$ | $6.830416344004197615 \times 10^{-2}$ |
| $a_3(k)$ | $8.289102927104288199 \times 10^{-1}$ | $5.802372048973001295 \times 10^{-1}$ | $3.315641170841715502 \times 10^{-1}$ |
| $a_4(k)$ | $8.846579905691620560e \times 10^{-2}$ | $4.423289952845810280 \times 10^{-2}$ | $2.653973971707486099 \times 10^{-2}$ |
| $a_5(k)$ | $8.562817629518102436 \times 10^{-1}$ | $5.137690577710861684 \times 10^{-1}$ | $2.568845288855430842 \times 10^{-1}$ |
| $a_6(k)$ | $1.630296867565628194 \times 10^{-1}$ | $1.141207807295939597 \times 10^{-1}$ | $4.890890602696884581 \times 10^{-2}$ |

Applying this,

$$
\mathbb{P}\left( \|\mathbf{W}_{i \cdot C} - \widehat{\mathbf{W}}_{i \cdot C}^{(T,L)}\| > \epsilon \right) \leq \mathbb{P}\left( \|\Phi_{CC} - \widehat{\Phi}_{CC}^{(T,L)}\| > \frac{\epsilon}{M+1} \right)
$$

$$
\leq \exp\left( -\frac{\epsilon^2 (T-L)}{32(2L+1)M^2(M+1)^2} + 6n \right)
$$

$$
\leq \exp\left( -\frac{\epsilon^2 (T-L)}{32(2L+1)(M+1)^4} + 6n \right). \tag{9}
$$

Thus, an upper bound on the sample complexity is

$$
\exp\left( -\frac{\epsilon^2 (T-L)}{32(2L+1)(M+1)^4} + 6n \right) = \delta
$$

$$
\implies \frac{\epsilon^2 (T-L)}{32(2L+1)(M+1)^4} = (\log(\delta) - 6n)
$$

$$
\implies (T-L) \approx O\left( \frac{(\log(\delta) - 6n)\left( (2L+1)M^4 \right)}{\epsilon^2} \right).
$$

For Bartlett estimator,

**Lemma B.2** (Veedu et al. (2024)). *Consider a linear dynamical system governed by equation 2. Suppose that the auto-correlation function $R_x(k) := \mathbb{E}\{X(n)X^\top(n+k)\}$ satisfies exponential decay, $\|R_x(k)\|_2 \leq C\delta^{-|k|}$ and that there exists $M$ such that $\frac{1}{M} \leq \lambda_{min}(\Phi_\mathbf{X}) \leq \lambda_{max}(\Phi_\mathbf{X}) \leq M$. Let $0 < \varepsilon_1, \varepsilon_2 < \varepsilon$ be such that $\varepsilon_2 = \varepsilon - \varepsilon_1$. Suppose that $N > \frac{2C\rho^{-1}}{(1-\rho^{-1})^2 \varepsilon_1}$, where $0 < \varepsilon_1$. Then $\forall \omega \in \Omega$,*

$$
\mathbb{P}\left( \|\widehat{\Phi}_{CC} - \Phi_{CC}\| > \epsilon \right) \leq \exp\left( -\frac{\epsilon^2 T}{128 N M^2} + 6n \right).
$$

Then,

$$
\mathbb{P}\left( \|\mathbf{W}_{i \cdot C} - \widehat{\mathbf{W}}_{i \cdot C}\| > \epsilon \right) \leq \mathbb{P}\left( \|\Phi_{CC} - \widehat{\Phi}_{CC}\| > \frac{\epsilon}{M+1} \right)
$$

$$
\leq \exp\left( -\frac{\epsilon^2 T}{128 N (M+1)^2 M^2} + 6n \right)
$$

$$
\leq \exp\left( -\frac{\epsilon^2 T}{128 N (M+1)^4} + 6n \right). \tag{10}
$$

## C  EXPERIMENTS

In this section, we provide the details on data generation, threshold tuning, and simulation results on statistical significance.

### C.1  DATA GENERATION

For MEG reconstruction of random networks, we performed the experiments over 25 random networks with $n = 6$ nodes. For each network iteration, a random DAG is first generated with at most

2 parents for each node. In this graph, the data is generated using the AR model, with the non-zero entries that respect the graph structure of the random DAG generated, using

$$x_i(t) + a_i(1)x_i(t-1) + a_i(2)x_i(t-2) + a_i(3)x_i(t-3) = \sum_{j \neq i} b_{ij}x_j(t-1) + e_i(t), \quad (11)$$

where $e_i(t)$ is zero mean i.i.d. Gaussian noise with diagonal covariance matrix and $i = 1, \ldots, n$. The data is generated continuously according to the AR model in equation 11. That is, for $t = 0$ to $T$, $\mathbf{X}(t)$ is computed based on equation 11, which satisfies Assumptions 1 and 2. Hence Lemma 4.2 and Algorithm 2 can be applied to reconstruct the essential graph.

The coefficients $a_i(\cdot)$ used for the data generation are shown in Table 3. It is to be noted that that for the random networks the $b_{ij}$ coefficients were generated randomly from $unif[0.2, 0.4]$. The coefficients are chosen this way to stop the time-series from becoming unbounded. In IID exogenous noise setting, $e_i(t) \sim N(0, 1)$ is used. Given the coefficients, the data is generated according to equation 11 for every $i \in V$.

## C.2 RANDOM NETWORK RECONSTRUCTION FROM DATA

The datasets for the 25 random DAGs were used in FD based W-PC, TD based W-PC, and Wiener-phase algorithm. The average error rate and computation plots are shown in Fig. 1(a) and 1(b). In this section, we quantify the performance of the algorithms statistically. The box and whisker plot of error rates for the 25 random networks are shown in Fig 4. As expected the error reduces as the number of samples increases.

As mentioned in Section 6 the performance of the proposed algorithms in this article depend on the choice of the thresholds. To tune the thresholds we use a sparsity based tuning approach. This approach uses bounds on the average, maximum, and minimum degree of the reconstructed topology to choose a threshold. The algorithm is shown in Algorithm 3 Such sparsity related information are typically be available in practice for DAG models.

---

**Algorithm 3** Threshold tuning-based on sparsity metric

**Input:** Data, upper limit of average degree $d_u$, lower limit of average degree $d_l$, maximum degree $d_{max}$, minimum degree $d_{min}$, lower limit of threshold $\tau_l$, upper limit of threshold $\tau_u$, resolution for tuning $\zeta$.
**Output:** Optimal choice of threshold $\widehat{\tau}$

1. Initialize threshold $\tau \leftarrow \tau_l$, $\widehat{\tau} \leftarrow \tau_l$
2. $p \leftarrow floor\left[(\tau_u - \tau_l)/\zeta\right]$
3. For $i = 1, \ldots, p$
   (a) Estimate $\widehat{G}$ using reconstruction algorithm and $\tau$
   (b) Compute average degree $(\hat{d})$, maximum degree $(\hat{d}_{max})$, minimum degree $(\hat{d}_{min})$ for skeleton of $\widehat{G}$.
   (c) If $d_l < \hat{d} < d_u$ and $\hat{d}_{max} < d_{max}$ and $\hat{d}_{min} > d_{min}$
      i. $\widehat{\tau} \leftarrow \tau$
      ii. break.
   (d) $\tau \leftarrow (\tau + \zeta)$
4. Return $\widehat{\tau}$

---

## C.3 DETAILS OF 20 NODES NETWORK

The 20 nodes network used in this article is a directed acyclic graph shown in Fig. 5. The network consists of 20 vertices, and 22 edges, with maximum and minimum degree of the nodes being 5 and 1. The maximum number of incoming edges at any specific node is 2 and maximum number of outgoing edges from any specific node is 3. As discussed earlier the data generative model for the 20 nodes network was taken to be of the form in equation 11. The $a_i(\cdot)$ coefficients for the 20 nodes network were chosen as shown in Table 4.

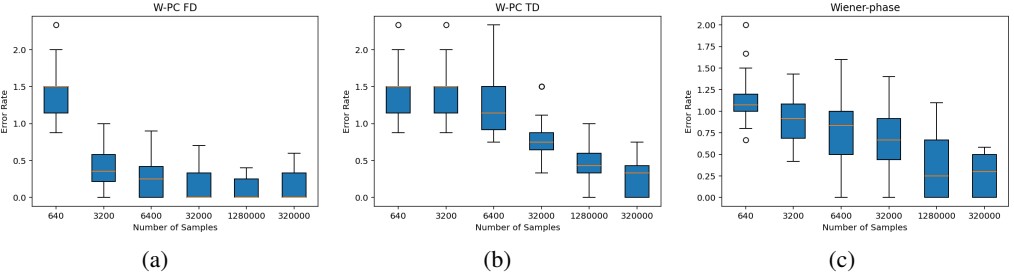

Figure 4: Box and whisker plots showing statistical performance of the (a) FD based W-PC (b) TD based W-PC (c) Wiener-Phase over 25 networks for the results presented in Section 6

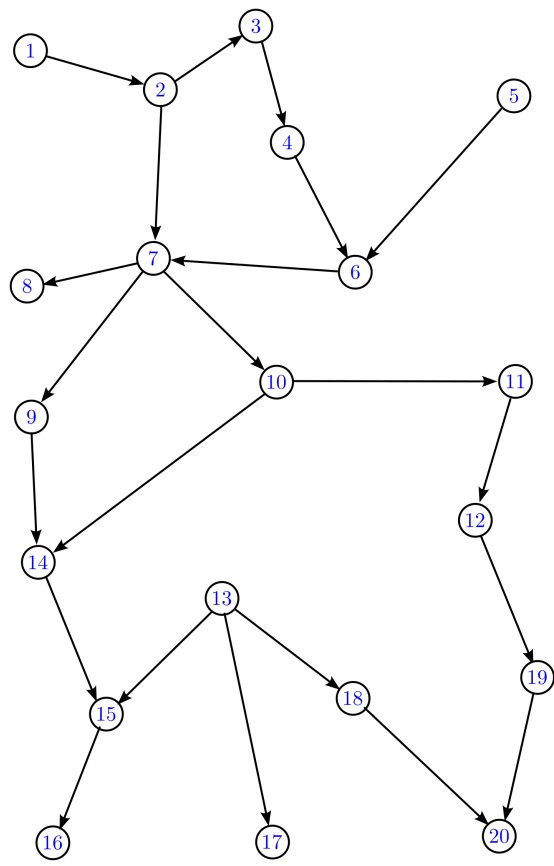

Figure 5: Generative graph of 20 nodes network

## C.4 DETAILS OF 50 NODES NETWORK

The 50 nodes network used in this article is a directed acyclic graph shown in Fig. 6. The network consists of 50 vertices, and 54 edges, with maximum and minimum degree of the nodes being 5 and 1. The maximum number of incoming edges at any specific node is 2 and maximum number of outgoing edges from any specific node is 3. The $a_i(\cdot)$ coefficients for the 50 nodes network were chosen as shown in Table 5.

Table 4: Coefficients used in data generation for 20 nodes network

|  | $k = 1$ | $k = 2$ | $k = 3$ |
|---|---|---|---|
| $a_1(k)$ | $1.739381261594696859e-02$ | $1.043628756956818150e-02$ | $5.218143784784090751e-03$ |
| $a_2(k)$ | $3.415208172002098808e-01$ | $1.366083268800839523e-01$ | $6.830416344004197615e-02$ |
| $a_3(k)$ | $8.289102927104288199e-01$ | $5.802372048973001295e-01$ | $3.315641170841715502e-01$ |
| $a_4(k)$ | $8.846579905691620560e-02$ | $4.423289952845810280e-02$ | $2.653973971707486099e-02$ |
| $a_5(k)$ | $8.562817629518102436e-01$ | $5.137690577710861684e-01$ | $2.568845288855430842e-01$ |
| $a_6(k)$ | $1.630296867565628194e-01$ | $1.141207807295939597e-01$ | $4.890890602696884581e-02$ |
| $a_7(k)$ | $1.739381261594696859e-02$ | $1.043628756956818150e-02$ | $5.218143784784090751e-03$ |
| $a_8(k)$ | $3.415208172002098808e-01$ | $1.366083268800839523e-01$ | $6.830416344004197615e-02$ |
| $a_9(k)$ | $8.289102927104288199e-01$ | $5.802372048973001295e-01$ | $3.315641170841715502e-01$ |
| $a_{10}(k)$ | $8.846579905691620560e-02$ | $4.423289952845810280e-02$ | $2.653973971707486099e-02$ |
| $a_{11}(k)$ | $8.562817629518102436e-01$ | $5.137690577710861684e-01$ | $2.568845288855430842e-01$ |
| $a_{12}(k)$ | $1.630296867565628194e-01$ | $1.141207807295939597e-01$ | $4.890890602696884581e-02$ |
| $a_{13}(k)$ | $1.739381261594696859e-02$ | $1.043628756956818150e-02$ | $5.218143784784090751e-03$ |
| $a_{14}(k)$ | $3.415208172002098808e-01$ | $1.366083268800839523e-01$ | $6.830416344004197615e-02$ |
| $a_{15}(k)$ | $8.289102927104288199e-01$ | $5.802372048973001295e-01$ | $3.315641170841715502e-01$ |
| $a_{16}(k)$ | $8.846579905691620560e-02$ | $4.423289952845810280e-02$ | $2.653973971707486099e-02$ |
| $a_{17}(k)$ | $8.562817629518102436e-01$ | $5.137690577710861684e-01$ | $2.568845288855430842e-01$ |
| $a_{18}(k)$ | $1.630296867565628194e-01$ | $1.141207807295939597e-01$ | $4.890890602696884581e-02$ |
| $a_{19}(k)$ | $1.739381261594696859e-02$ | $1.043628756956818150e-02$ | $5.218143784784090751e-03$ |
| $a_{20}(k)$ | $3.415208172002098808e-01$ | $1.366083268800839523e-01$ | $6.830416344004197615e-02$ |

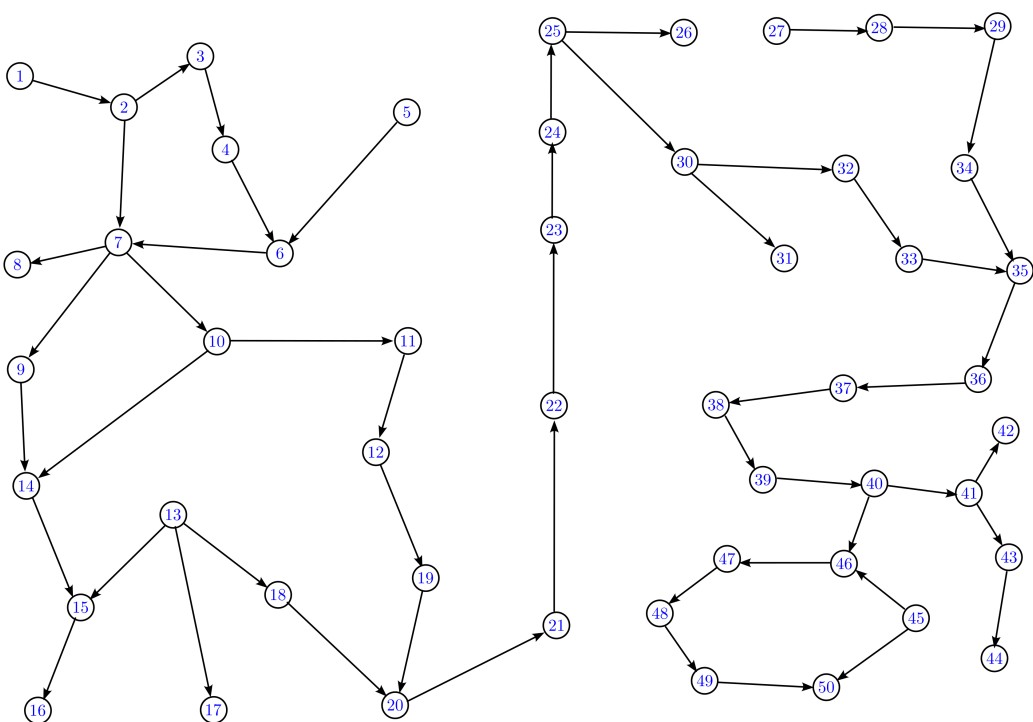

Figure 6: Generative graph of 50 nodes network

## C.5 ROBUSTNESS OF WIENER PHASE ALGORITHM TO PHASE CONDITION VIOLATION

In addition to the results in the main article, our algorithm remains robust to assumption violation as demonstrated through example here. To study the robustness of the Wiener Phase algorithm we employed the Wiener phase algorithm on synthetic data that violates the phase alignment assumption. The model was taken to be of the following form: $a_{i,0}x_i(t) + a_{i,1}x_i(t-1) + a_{i,2}x_i(t-2) = \sum_{j \neq i}(b_{ij,0}x_j(t-1) + b_{ij,1}x_j(t-2) + b_{ij,2}x_j(t-3)) + e_i(t)$, where the coefficients $b_{ij,k}$ were chosen randomly from $[0.1, 0.9]$, and $a_{i,k}$ were chosen such that the time-series remain bounded. The network was taken to be a 20 nodes network shown in Appendix C.3. For the 20-node network with 22 edges, and an N=32 point FFT, and threshold of $0.15$ the results are shown in Table 6.

Table 5: Coefficients used in data generation for 50 nodes network

|  | $k = 1$ | $k = 2$ | $k = 3$ |
|---|---|---|---|
| $a_1(k)$ | $1.739381261594696859e-02$ | $1.043628756956818150e-02$ | $5.218143784784090751e-03$ |
| $a_2(k)$ | $3.415208172002098808e-01$ | $1.366083268800839523e-01$ | $6.830416344004197615e-02$ |
| $a_3(k)$ | $8.289102927104288199e-01$ | $5.802372048973001295e-01$ | $3.315641170841715502e-01$ |
| $a_4(k)$ | $8.846579905691620560e-02$ | $4.423289952845810280e-02$ | $2.653973971707486099e-02$ |
| $a_5(k)$ | $8.562817629518102436e-01$ | $5.137690577710861684e-01$ | $2.568845288855430842e-01$ |
| $a_6(k)$ | $1.630296867565628194e-01$ | $1.141207807295939597e-01$ | $4.890890602696884581e-02$ |
| $a_7(k)$ | $1.739381261594696859e-02$ | $1.043628756956818150e-02$ | $5.218143784784090751e-03$ |
| $a_8(k)$ | $3.415208172002098808e-01$ | $1.366083268800839523e-01$ | $6.830416344004197615e-02$ |
| $a_9(k)$ | $8.289102927104288199e-01$ | $5.802372048973001295e-01$ | $3.315641170841715502e-01$ |
| $a_{10}(k)$ | $8.846579905691620560e-02$ | $4.423289952845810280e-02$ | $2.653973971707486099e-02$ |
| $a_{11}(k)$ | $8.562817629518102436e-01$ | $5.137690577710861684e-01$ | $2.568845288855430842e-01$ |
| $a_{12}(k)$ | $1.630296867565628194e-01$ | $1.141207807295939597e-01$ | $4.890890602696884581e-02$ |
| $a_{13}(k)$ | $1.739381261594696859e-02$ | $1.043628756956818150e-02$ | $5.218143784784090751e-03$ |
| $a_{14}(k)$ | $3.415208172002098808e-01$ | $1.366083268800839523e-01$ | $6.830416344004197615e-02$ |
| $a_{15}(k)$ | $8.289102927104288199e-01$ | $5.802372048973001295e-01$ | $3.315641170841715502e-01$ |
| $a_{16}(k)$ | $8.846579905691620560e-02$ | $4.423289952845810280e-02$ | $2.653973971707486099e-02$ |
| $a_{17}(k)$ | $8.562817629518102436e-01$ | $5.137690577710861684e-01$ | $2.568845288855430842e-01$ |
| $a_{18}(k)$ | $1.630296867565628194e-01$ | $1.141207807295939597e-01$ | $4.890890602696884581e-02$ |
| $a_{19}(k)$ | $1.739381261594696859e-02$ | $1.043628756956818150e-02$ | $5.218143784784090751e-03$ |
| $a_{20}(k)$ | $3.415208172002098808e-01$ | $1.366083268800839523e-01$ | $6.830416344004197615e-02$ |
| $a_{21}(k)$ | $1.739381261594696859e-02$ | $1.043628756956818150e-02$ | $5.218143784784090751e-03$ |
| $a_{22}(k)$ | $3.415208172002098808e-01$ | $1.366083268800839523e-01$ | $6.830416344004197615e-02$ |
| $a_{23}(k)$ | $8.289102927104288199e-01$ | $5.802372048973001295e-01$ | $3.315641170841715502e-01$ |
| $a_{24}(k)$ | $8.846579905691620560e-02$ | $4.423289952845810280e-02$ | $2.653973971707486099e-02$ |
| $a_{25}(k)$ | $8.562817629518102436e-01$ | $5.137690577710861684e-01$ | $2.568845288855430842e-01$ |
| $a_{26}(k)$ | $1.630296867565628194e-01$ | $1.141207807295939597e-01$ | $4.890890602696884581e-02$ |
| $a_{27}(k)$ | $1.739381261594696859e-02$ | $1.043628756956818150e-02$ | $5.218143784784090751e-03$ |
| $a_{28}(k)$ | $3.415208172002098808e-01$ | $1.366083268800839523e-01$ | $6.830416344004197615e-02$ |
| $a_{29}(k)$ | $8.289102927104288199e-01$ | $5.802372048973001295e-01$ | $3.315641170841715502e-01$ |
| $a_{30}(k)$ | $8.846579905691620560e-02$ | $4.423289952845810280e-02$ | $2.653973971707486099e-02$ |
| $a_{31}(k)$ | $8.562817629518102436e-01$ | $5.137690577710861684e-01$ | $2.568845288855430842e-01$ |
| $a_{32}(k)$ | $1.630296867565628194e-01$ | $1.141207807295939597e-01$ | $4.890890602696884581e-02$ |
| $a_{33}(k)$ | $1.739381261594696859e-02$ | $1.043628756956818150e-02$ | $5.218143784784090751e-03$ |
| $a_{34}(k)$ | $3.415208172002098808e-01$ | $1.366083268800839523e-01$ | $6.830416344004197615e-02$ |
| $a_{35}(k)$ | $8.289102927104288199e-01$ | $5.802372048973001295e-01$ | $3.315641170841715502e-01$ |
| $a_{36}(k)$ | $8.846579905691620560e-02$ | $4.423289952845810280e-02$ | $2.653973971707486099e-02$ |
| $a_{37}(k)$ | $8.562817629518102436e-01$ | $5.137690577710861684e-01$ | $2.568845288855430842e-01$ |
| $a_{38}(k)$ | $1.630296867565628194e-01$ | $1.141207807295939597e-01$ | $4.890890602696884581e-02$ |
| $a_{39}(k)$ | $1.739381261594696859e-02$ | $1.043628756956818150e-02$ | $5.218143784784090751e-03$ |
| $a_{40}(k)$ | $3.415208172002098808e-01$ | $1.366083268800839523e-01$ | $6.830416344004197615e-02$ |
| $a_{41}(k)$ | $8.562817629518102436e-01$ | $5.137690577710861684e-01$ | $2.568845288855430842e-01$ |
| $a_{42}(k)$ | $1.630296867565628194e-01$ | $1.141207807295939597e-01$ | $4.890890602696884581e-02$ |
| $a_{43}(k)$ | $1.739381261594696859e-02$ | $1.043628756956818150e-02$ | $5.218143784784090751e-03$ |
| $a_{44}(k)$ | $3.415208172002098808e-01$ | $1.366083268800839523e-01$ | $6.830416344004197615e-02$ |
| $a_{45}(k)$ | $8.289102927104288199e-01$ | $5.802372048973001295e-01$ | $3.315641170841715502e-01$ |
| $a_{46}(k)$ | $8.846579905691620560e-02$ | $4.423289952845810280e-02$ | $2.653973971707486099e-02$ |
| $a_{47}(k)$ | $8.562817629518102436e-01$ | $5.137690577710861684e-01$ | $2.568845288855430842e-01$ |
| $a_{48}(k)$ | $1.630296867565628194e-01$ | $1.141207807295939597e-01$ | $4.890890602696884581e-02$ |
| $a_{49}(k)$ | $1.739381261594696859e-02$ | $1.043628756956818150e-02$ | $5.218143784784090751e-03$ |
| $a_{50}(k)$ | $3.415208172002098808e-01$ | $1.366083268800839523e-01$ | $6.830416344004197615e-02$ |

Table 6: Performance of Wiener Phase Algorithm on Phase Violation

| Sample Count | $CS\%$ | $TPR\%$ | $(1-FPR)\%$ | Run Time (s) |
|---|---|---|---|---|
| 6400 | 86.52 | 87.09 | 99.43 | 0.117 |
| 32000 | 88.89 | 90.32 | 98.57 | 0.2298 |
| 128000 | 92.69 | 93.55 | 99.14 | 1.279 |

It can be observed from the above results that even though the phase alignment assumption is violated the Wiener phase algorithm produces highly accurate results. This shows that the algorithm is robust to the assumption being violated.

## C.6  APPLICATIONS SATISFYING ASSUMPTION 1

Any network whose states evolve according to the following dynamics,

$$\sum_{m=1}^{l} a_{m,i} \frac{\mathrm{d}^m x_i}{\mathrm{d}t^m} + a_{0,i} x_i = \sum_{j=1}^{n} c_{ij}(x_j - x_i) + p_i(t),$$

satisfies Assumption 1. Some examples include Talukdar et al. (2020)

1. **Consensus dynamics**: In multi-agent systems, distributed decision making often follows the following first order consensus dynamics, where each agent $i$ updates its states according to

$$\frac{\mathrm{d}x_i}{\mathrm{d}t} = \sum_{j=1}^{n} c_{ij}(x_j - x_i) + p_i(t),$$

where $p_j$ denotes the receiver noise for agent $j$ Siami et al. (2017). Identifying the communication topology of a network of agents in a multi-agent system is a relevant objective of a cyber attacker, and appropriate hardware/software tools need to be designed to avert such attacks.

2. **Thermal Resistance Capacitance (RC) networks**: One of the popular approaches in modeling the real-time control of building is the gray box modeling with lumped parameters. Here, the discretized physical spaces are modeled as RC models, where each discretized physical space is represented by a common temperature. An example lumped parameter model is

$$C_i \frac{\mathrm{d}T_i}{\mathrm{d}t} = \sum_{j=1}^{n} \frac{T_j - T_i}{R_{ij}} + p_j(t),$$

where $T_i$ denotes the temperature, $C_i > 0$ denotes the capacitance and $p_j$ denotes the total internal heat generated of zone $j$. $R_{ij} \geq 0$ is the thermal resistance between zones $i$ and $j$.

3. **Power Grid Dynamics**: When the disturbances in the power grid are small, the deviation in voltage phase angle dynamics at bus $j$, denoted by $\theta_j$, can be modeled by a swing equation, for example,

$$M_i \frac{\mathrm{d}^2 x_i}{\mathrm{d}t^2} + D_i \frac{\mathrm{d}x_i}{\mathrm{d}t} = \sum_{j=1, j \neq i}^{n} c_{ij}(x_j - x_i) + p_j,$$

where $p_j$ denotes the receiver noise for agent $j$.

4. **Linearized Chemical Reaction networks:** Chemical engineering processes often operate in transient states and are generally governed by ordinary differential equations (ODEs). ODEs are used in applications such as chemical stirring tank design, heat exchangers, or biological cell growth. Example, see here.

## C.7  EXAMPLE WHERE GRANGER CAUSALITY FAILS

An essential assumption for applying Granger causality is that the underlying influence dynamics should be temporally strict causal. That is, if $x$ influences $y$, then "only" the "past" values of $x$ in time should influence $y$. Granger causality will fail if the present value of $y$ depends on the present value of $x$ as well. However, our approach can accommodate any linear dynamical system where the present value of $y$ can depend on present, past, or future values of $x$. We generated a 6-node synthetic network with zero lag dependency between the nodes that follow the dynamics described as below

$$x_i(t) + a_i(1)x_i(t-1) + a_i(2)x_i(t-2) + a_i(3)x_i(t-3) = \sum_{\tau=0}^{3} \sum_{j \neq i} b_{ij} x_j(t-\tau) + e_i(t). \quad (12)$$

Table 7: Algorithm performance on non-linear system

| Algorithm | $TPR\%$ | $(1-FPR)\%$ | $CS\%$ |
|---|---|---|---|
| FFT-WPC | 100 | 100 | 100 |
| Wiener Phase | 100 | 100 | 100 |

The following results show that the GC fails to reconstruct the full network even at $10^6$ samples. Our algorithms achieve zero reconstruction error with fewer samples.

- Wiener Phase: perfect reconstruction at 6400 samples with a $N = 32$ point FFT and a threshold of $0.1$.
- FFT-WPC: perfect reconstruction at 15000 samples with a $N = 32$ point FFT and a threshold of $0.074$.
- Granger Causality: CS = $80\%$ at $3.2 \times 10^6$ samples, with significance level $\alpha = 0.086$.

### C.8 APPLICATION OF ALGORITHMS TO NON-LINEAR SYSTEMS

Note that the proposed algorithms are purely data driven and thus are applicable to any set of time-series data; the underlying data generative system does not determine the applicability of our algorithms. This is in contrast to other methods, for example, where the algorithm depends on a parametrized model, and the parameters are determined via data. To demonstrate that our algorithms may be effective in many non-linear system we present an example below.

We used a generative model of the form

$$a_{i,0}x_i(t) + a_{i,1}x_i(t-1) + a_{i,2}x_i(t-2)) + a_{i,3}x_i(t-3) = \sum_{j \neq i} b_{ij}(x_j(t) + 0.1x_j^2(t-1)) + e_i(t), \tag{13}$$

where $e_i(t)$ is standard Gaussian noise. $b_{ij}$ are uniform random numbers, and $a_i$ are chosen such that the time series remains bounded (poles of the transfer functions remain within the unit circle). The above model was used to generate data for the 20 nodes network of Appendix C.3. The generated data was then used in the FFT based Wiener PC and the Wiener Phase algorithm with thresholds $0.04$ and $N = 32$ point FFT in both algorithms. The results are as shown in Table 7.

It can be observed that the algorithm produce almost perfect reconstruction results for the nonlinear system described above.

### C.9 DESCRIPTION OF SETUP FOR EXPERIMENTS ON CIRCUITS

As described in Rana et al. (2025), the experimental setup for data acquisition from transistor based circuits that was described in Section 6.4 consists of the following hardware sub components:

- a computer station with LabVIEW 2021,
- an NI-cRIO 9054 equipped with Xilinx Artix-7 A100T FPGA,
- an NI-9223 module with 4 $\pm 10$V isolated analog input channels, 16-bit simultaneous sampling of all channels at maximum 1MS/s rate,
- a 600W Multi-Range 150V/10A bench-top DC Power Supply (Model 9206) from BK Precision,
- a 12V 50W AC/DC Converter (LRS-50-12) from Mean Well USA Inc,
- low noise instrumentation amplifier (AD8421) based custom made measurement circuits,
- hardware under test (HUT) which was either a BJT or MOSFET amplifier circuit. The PCB implementations of the hardware under tests are shown in Fig.2 and Fig. 3.

The circuit under test were energized using the bench-top DC supply from BK Precision and the NI-cRIO DAQ system was power from the 12V AC/DC converter from Mean Well USA. The node voltages, $\{v_1,\ v_2,\ v_3,\ v_4\}$, as shown in Fig. 2 and Fig. 3 were then first amplified using the AD8421

Table 8: Comparison of algorithms on simulated analog electronic circuit dataset

| Algorithm | $F_1$ | $CS$ | $TPR$ | $\widetilde{FPR}$ | Run Time |
|---|---|---|---|---|---|
| Fisher-Z PC | 37.84 | 37.74 | 70 | 67.74 | 0.1897 |
| CD-NOD | 37.84 | 37.74 | 70 | 67.74 | 0.298 |
| GC | 80 | 91.94 | 100 | 91.94 | 0.0928 |
| TPC | 60.61 | 79.03 | 100 | 79.03 | 97.84 |
| FFT-WPC | 100 | 100 | 100 | 100 | 10.97 |
| Wiener Phase | 90.91 | 96.8 | 100 | 96.8 | 0.153 |

instrumentation amplifier. The amplified signals were recorded using the NI-cRio and NI-9223 based data acquisition system. The LabVIEW software was used to deploy the data acquisition algorithms required for the NI-cRio. A screen capture of a segment of the recorded voltage measurements are shown in Fig. 7.

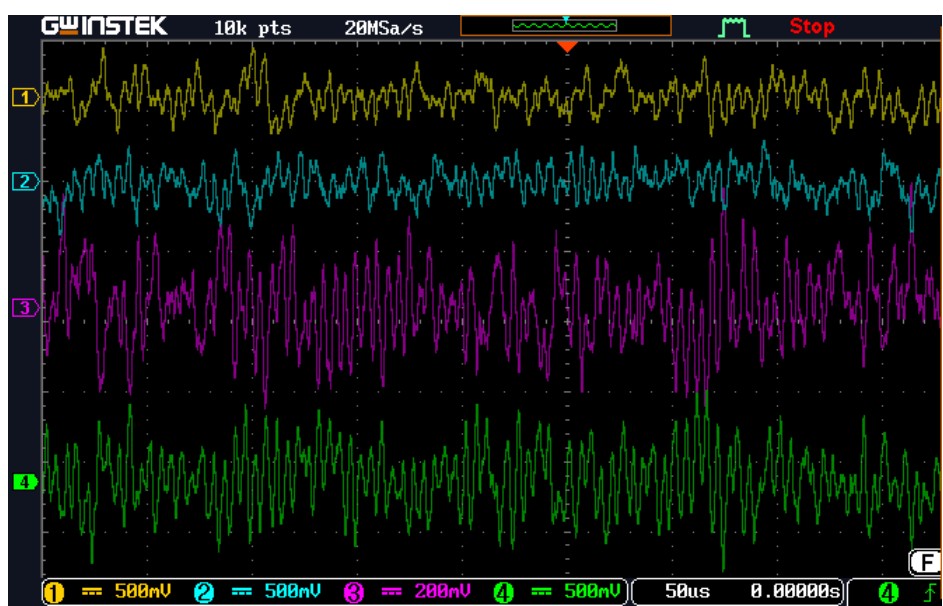

Figure 7: Waveforms of recorded voltage measurements for the experiments on physical hardware circuits displayed on an oscilloscope.

### C.10 RESULTS ON LARGER CIRCUIT SIMULATED IN CADENCE

In this section, we demonstrate the application of the proposed methods in larger circuits that employ transistors, capacitors, and resistors. To demonstrate the effectiveness of the proposed algorithms in this article over state-of-the-art methods on larger circuits, we collected a dataset containing high fidelity simulation data. A schematic of the circuit simulated in Cadence virtuoso, which is a high fidelity circuit simulator, is shown in Fig. 8. A spice model of a commercial bipolar junction transistor (BJT) (MMBT2222ATT1G) was used for the simulation. The circuit contains multiple transistor amplifier stages connected in common emitter configuration with bypass capacitors at the emitter. Also, the different stages are connected through capacitors leading to dynamic nature of the circuit. A total of 799999 samples were collected for each of the node voltages $\{v_1, ... v_9\}$ shown in Fig. 8. The comparative results using the data collected from the simulated dynamic circuit is shown in Table 8. It can be observed that our proposed approach outperforms state-of-the-art methods. Note that the FFT-WPC and the Wiener-Phase produce almost perfect reconstruction, and the Wiener-Phase algorithm is the fastest. The FFT-WPC achieves the highest F1 score followed by Wiener-Phase, GC, TPC, CD-NOD, and Fisher-Z PC in descending order.

### C.11 EFFECT OF FREQUENCY DISCRETIZATION ON ALGORITHM PERFORMANCE AND CHOICE OF $N$

In this section, we present results showing the effects of the frequency discretization and sampling on the proposed algorithm. As described earlier, to compute the FD based Wiener filters we sample

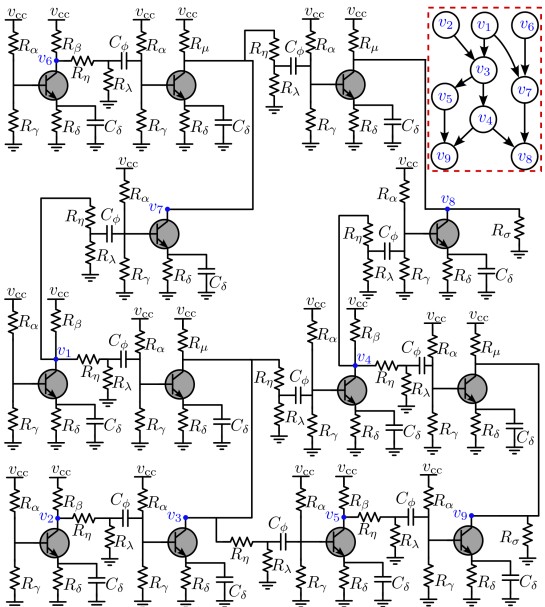

Figure 8: Schematic and generative graph of simulated transistor amplifier
Rana et al. (2025)

the frequency interval $\Omega = [0, 2\pi]$ at $N$ evenly spaced points. To quantify the effects of frequency discretization and sampling, we evaluated the performance metrics of the algorithms at different values of $N$. The data generative model of equation 11 used for generating data for 25 random DAGs each containing 10 nodes and randomly generated edges. The datasets, each containing 128000 samples, were used in Algorithms 1 and 2 with the sparsity based threshold tuning Algorithm 3 for different values of $N$. The average error rates and $F_1$ scores over the 25 random DAGs are shown in Fig. 9. It can be observed that the algorithms perform better as the value of $N$ increases. However, note that for $N \geq 32$ the improvement in algorithm performance with increasing $N$ is incremental. Although increasing the value of $N$ may increase the performance beyond 32 for the presented example, it would also increase the computational burden as the complexity of the algorithms is $O(\log N)$. Therefore, to maintain an acceptable tradeoff between complexity and performance we choose $N = 32$.

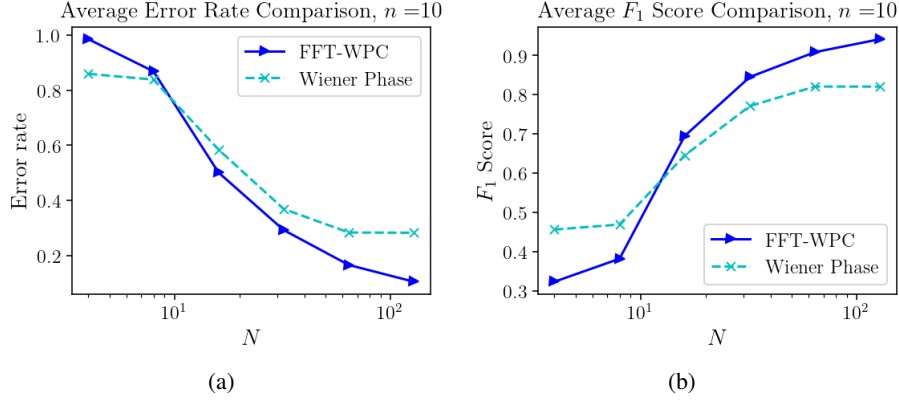

Figure 9: (a) Average error rate and (b) $F_1$ score for different values of $N$ for FD Wiener PC and Wiener-Phase over 25 random networks

## D   FREQUENCY DISCRETIZATION

The LDIM in equation 2 can be sampled at frequencies $\omega$ in $\Omega_N = \{\omega_0, \ldots, \omega_{N-1}\}$, where $\omega_k := \frac{2\pi k}{N}$, thus converting the continuous frequency model to a practical model involving a finite set of frequencies. In TD this is equivalent to the finite impulse response model

$$\mathbf{x}(n) = \sum_{l=0}^{N-1} \mathbf{h}(l)\mathbf{x}(n-l) + \mathbf{e}(n), \tag{14}$$

where $n = 0, \ldots, N-1$, $\mathbf{x}(n)$ and $\mathbf{e}(n)$ are periodic with period $N$, and $\mathbf{h}(l)$ is non-zero for at most $N-1$ lags. For the circular convolution model equation 14, $\mathbf{X}(\omega_k) := \frac{1}{\sqrt{N}} \sum_{n=0}^{N-1} \mathbf{x}(n) e^{-2\pi k n/N}$, $k = 0, \ldots, N-1$, provides the Discrete Fourier Transform (DFT) of the sequence $\mathbf{x}(n)$. Similar DFT expressions hold for $\mathbf{H}(\omega_k)$ and $\mathbf{E}(\omega_k)$ as well, assuming $\mathbf{h}(n)$ and $\mathbf{e}(n)$ are periodic. The directed graph associated with equation 14, $\breve{G} = (V, \breve{E})$, is defined such that the edge $u \to v$ exists in $\breve{E}$ if $\mathbf{H}_{vu}(\omega_k) \neq 0$ for some $k$. It is well known that the Fourier coefficients can be approximated via discretization Grafakos (2008). A uniform and explicit convergence rate for signals with bounded variation is provided below in Section D.1 that shows a convergence rate of $1/\sqrt{N}$ for approximating equation 1 with equation 14.

### D.1   CONVERGENCE OF DFT TO DTFT

Notation: For a continuous function $f$, $\widehat{f}$ denotes the Fourier transform of $f$.

**Proof**:

*Definition* 1. Grafakos (2008) The total variation of a continuous complex-valued function $f$, defined on $[a, b] \subset \mathbb{R}$ is the quantity, $V_a^b(f) := \sup_{p \in \mathcal{P}} \sum_{i=0}^{n_p-1} |f(x_{i+1} - f(x_i)|$, where $P = \{x_0, \ldots, x_{n_p-1}\}$ is a partition and $\mathcal{P}$ is the set of all partitions of $[a, b]$.

A continuous complex-valued function $f$ is said to be of bounded variation on a chosen interval $[a, b] \subset \mathbb{R}$ if its total variation is finite, i.e. if $V_a^b(f) < +\infty$.

**Lemma D.1.** *For a function $\widehat{f}$ of bounded variation $V$, on $[0, 2\pi]$ and $N \geq 1$, the estimation of $f(n) = \int_0^{2\pi} \widehat{f}(\omega)e^{j\omega n}d\omega$ given by $f^{(N)}(n) = \frac{1}{N}\sum_{k=0}^{N-1} e^{j2\pi nk/N} \widehat{f}(2\pi k/N)$ for $|n| \leq \sqrt{N}$ and zero otherwise, satisfies $\|f - f^{(N)}\|_{\ell_\infty(\Omega)} \leq C/\sqrt{N}$.*

*Proposition* 1. For $\widehat{f}$ with bounded variation, $V_a^b(\widehat{f})$, on $[a, b]$ and $t \in [a, b]$

$$\left| \int_a^b \widehat{f}(x)dx - (b-a)\widehat{f}(t) \right| \leq (b-a)V_a^b(\widehat{f}).$$

**Proof:** Since $\widehat{f}$ has bounded variation, $M := \sup_{x \in [a,b]} \widehat{f}(x)$ and $m := \inf_{x \in [a,b]} \widehat{f}(x)$ are both finite. Thus,

$$\left| \int_a^b \widehat{f}(x)dx - (b-a)\widehat{f}(t) \right| \leq \int_a^b \left| \widehat{f}(x) - \widehat{f}(t) \right| dx \leq (M-m)(b-a) \leq (b-a)V_a^b(\widehat{f}).$$

*Corollary* 1. For $\widehat{f}$ a function of bounded variation on $[0, 1]$, given by $V_0^1(\widehat{f})$, the error in a Riemann sum approximation,

$$\varepsilon_N := \left| \int_0^1 \widehat{f}(x)dx - \frac{1}{N}\sum_{n=0}^{N-1} \widehat{f}(n/N) \right| \leq V_0^1(\widehat{f})/N.$$

**Proof:** Note that

$$\varepsilon_N \leq \sum_{n=0}^{N-1} \left| \int_{n/N}^{n+1/N} \widehat{f}(x)dx - \frac{1}{N}\widehat{f}(n/N) \right| \leq \sum_{n=0}^{N-1} V_{n/N}^{(n+1)/N}(\widehat{f})/N \leq V_0^1(\widehat{f})/N,$$

where the second inequality follows from Proposition 1.

*Proposition* 2 (Proposition 3.2.14, Grafakos (2008)). If $\widehat{f}$ is of bounded variation, then

$$f(n) \leq \frac{V_0^1(\widehat{f})}{2\pi|n|}, \; n \neq 0.$$

*Corollary* 2. For $n \in \mathbb{N}$, and $\varepsilon > 0$ fixed. the estimation of the Fourier series $f(n)$ given by

$$f^{(N)}(n) = \begin{cases} \sum_{k=0}^{N-1} e^{j2\pi kn/N} \widehat{f}(k/N) & |n| \leq N^{1-\varepsilon} \\ 0 & \text{otherwise} \end{cases}$$

converges uniformly to $f(n)$.

**Proof:** For $|n| < N^{1-\varepsilon}$, it follows by Corollary 2 that

$$\left| f(n) - f^{(N)}(n) \right| \leq \sqrt{2} \frac{\|\widehat{f}\|_\infty 2\pi|n| + V_0^1(\widehat{f})}{N}$$
$$\leq \frac{\sqrt{2}\|\widehat{f}\|_\infty 2\pi}{N^\varepsilon} + \frac{\sqrt{2} V_0^1(\widehat{f})}{N}.$$

Similarly, for $n \geq N^{1-\varepsilon}$

$$\left| f(n) - f^{(N)}(n) \right| \leq \frac{V_0^1(\widehat{f})}{2\pi|n|} \leq \frac{V_0^1(\widehat{f})}{2\pi N^{1-\varepsilon}}.$$

The proof of Theorem D.1 follows by letting $\varepsilon = 1/2$.

$\blacksquare$

