# OpenReview forum: "Frequency-Domain Better than Time-Domain for Causal Structure Recovery in Dynamical Systems on Networks"
_ICLR.cc/2026/Conference — ICLR 2026 Poster_

### Official Review · Reviewer_eWah · 2025-10-25

**Soundness:** 2
**Presentation:** 2
**Contribution:** 3
**Rating:** 4
**Confidence:** 3

**Summary:**

The work compares time- versus frequency-domain estimation of multivariate Wiener filters for causal structure recovery under a linear dynamical influence model, proves computational gains via FFT, provides finite-sample concentration bounds for both domains, and proposes Wiener-PC (CI testing via Wiener coefficients) and Wiener-Phase (phase/imaginary-part–based skeleton and collider discovery), validated on synthetic and real river-runoff data to support the claims.

**Strengths:**

- S1: Uses the zero/nonzero support of multivariate Wiener coefficients as a principled CI surrogate, with FFT yielding explicit complexity gains over time-domain least squares, and finite-sample bounds provided in both domains.

- S2: Introduces Wiener-Phase, which leverages the imaginary part/phase of frequency-domain Wiener filters to recover the skeleton and identify colliders under phase assumptions, with strong empirical scalability.

- S3: Provides broad empirical evidence on synthetic node networks and a real-world dataset, showing superior accuracy and runtime trade-offs versus multiple baselines.

**Weaknesses:**

- W1: Key guarantees for Wiener‑Phase rely on structural assumptions (e.g., equal phase of all incoming edges and a nonzero imaginary part of transfer functions), which are somewhat strict.

- W2: The approach hinges on heuristic thresholding of Wiener coefficients, tuned via sparsity/degree constraints rather than statistically calibrated procedures, which may make results sensitive to sampling and frequency discretization.

- W3: Baseline coverage omits widely used frequency-domain causal discovery families such as spectral Granger variants, making it harder to position the proposed methods relative to established spectral measures beyond time-domain Granger.

**Questions:**

See Weaknesses

---

> ### Author Response · Authors · 2025-11-22
> **Part-1**
>
> **Reviewer Comment:** "Key guarantees for Wiener‑Phase rely on structural assumptions (e.g., equal phase of all incoming edges and a nonzero imaginary part of transfer functions), which are somewhat strict."
>
> **Author Response:**
>
> As acknowledged in the article, the assumptions may seem restrictive. However, the assumptions hold in many real-world applications such as, consensus dynamics, thermal RC networks, power grid dynamics, linearized chemical reactions, and many electronic circuits as listed in Appendix C.6. To further support our claim, we have now added results in the article showing the efficacy of the Wiener-Phase algorithm in real transistor-based circuits implemented on printed circuit boards. A summary of the results is provided below.
> The authors of [4] have shown that the relationships among the output voltages of different amplifier stages can be expressed in the modeling framework of our article here. Therefore, to apply our proposed method to electronic circuits we acquired datasets for a multistage transistor amplifier from high fidelity simulation in cadence virtuoso. The circuit consists of $13$ transistors and $9$ measurement points with resistive and capacitive elements connecting the various stages. For a detailed diagram of the circuit please see Section 6.4 of the updated article. A set of $9$ time-series measurements containing $799999$ samples each was obtained. For the given circuit we compared the reconstruction results of our proposed methods with that of the base line algorithms. The results are reported in the table below:
>
> | Algorithm    | F1 Score (%)   | CS (%) | TPR (%)  | 1-FPR (%) | Runtime (seconds) |
> | :---------------- |:-----: |:-----: |:------: | :--------: | :----------: |
> | Fisher-Z PC | 37.84  | 37.74 | 70 | 67.74 | 0.1897 |
> | GC       | 80 | 91.94 | 100 | 91.94 | 0.0928 |
> | TPC     | 60.61  | 79.03 | 100 | 79.03 | 97.84 |
> | CD-NOD | 37.84 | 37.74  | 70  | 67.74  | 0.298 |
> | FFT-WPC   | 100 | 100  | 100  | 100  | 10.97  |
> | Wiener Phase | 90.91 | 96.8  | 100  | 96.8  | 0.153  |
>
> As described earlier, the dataset was generated from a circuit contains dynamics. Note that the Wiener-Phase algorithm produces almost perfect reconstruction for the circuit.
> To further validate our methods in real-world applications we compared the proposed method with the SOTA methods using data acquired from physical hardware setup. We acquired a dataset containing $4$ time-series voltage measurements with $500000$ samples from the authors of [4]. The circuit which was implemented on a printed circuit board consists of a multistage transistor amplifier built using MMBT2222ATT1G bipolar junction transistor from Onsemi, resistors, and capacitors. To measure and acquire the data the authors of [4] used instrumentation amplifiers and Ni-cRio data acquisition system. The table below summarized the comparative results of reconstruction.
>
> | Algorithm    | F1 Score (%)   | CS (%) | TPR (%)  | 1-FPR (%) | Runtime (seconds) |
> | :---------------- |:-----: |:-----: |:------: | :--------: | :----------: |
> | Fisher-Z PC | 40  | 0 | 100 | 0 | 0.019 |
> | GC       | 75  | 77.78 | 100 | 77.78 | 0.0391 |
> | TPC     | 60  | 55.56  | 100 | 55.56 | 23.641 |
> | CD-NOD | 40  | 0  | 100  | 0  | 0.0309 |
> | FFT-WPC   | 100  | 100 | 100 | 100  | 0.0798 |
> | Wiener Phase | 100  | 100 | 100  | 100  | 0.0419 |
>
> Observe that the proposed method outperforms the SOTA methods in terms of both accuracy and speed. Moreover, the Wiener-Phase algorithm produces perfect reconstruction of the generative graph of the circuit implemented in the hardware. This shows that indeed, Wiener-Phase algorithm is effective in such an application domain.

---

> ### Author Response · Authors · 2025-11-22
> **Part-2**
>
> **Reviewer Comment:** "The approach hinges on heuristic thresholding of Wiener coefficients, tuned via sparsity/degree constraints rather than statistically calibrated procedures, which may make results sensitive to sampling and frequency discretization."
>
> **Author Response:**
>
> Even though our method of threshold tuning based on sparsity constraints is not statistically calibrated, the method was shown to be effective in the demonstrations. Since such constraints are available in many application domains the use of sparsity metric-based tuning provides a relatively straight-forward way to tune the algorithms. Below are some examples where the method can be effective:
>
> - Power System:  In power system networks different bus/nodes are connected together through electric wires which are the transmission lines. In practice the design infrastructure of power networks imposes constraints on the connectivity of the nodes due to different reasons such as cost of design and operation and geographical location. As a result, a particular node in a power network can be connected to only certain geographical neighbors leading to constraints on the sparsity of the connectivity matrix. Moreover, this information is typically available to the power system operator.
>
> - Circuits: Another example is analog electronic circuits. In a typical electronic amplifier network the connectivity between different stages is upper bounded due to design constraints and complexity. Such constraints on design lead to sparsity patterns in the networks. Such information is typically known to designers and in many cases can be inferred from the datasheets of devices.
>
> Although we have demonstrated the effectiveness of the sparsity based tuning method we acknowledge that a statically calibrated method could provide more general way of threshold tuning. However, given the vast scope of such study we believe such study should be part of future research.
>
> **Reviewer Comment:** "Baseline coverage omits widely used frequency-domain causal discovery families such as spectral Granger variants, making it harder to position the proposed methods relative to established spectral measures beyond time-domain Granger."
>
> **Author Response:**
>
> We would like to emphasize that the Granger causality algorithm implemented in the article does use frequency domain methods to estimate causal dependency. The granger causality algorithm was obtained from the open source Python library known as *TimeAwarePC* by the authors of [3]. The *TimeAwarePC* library uses the granger causality implementation of the *nitime* python repository which follows the method described in [7] for estimation of the causal effects in frequency domain. The Granger causality algorithm fits the time-series to a multivariate auto regressive model of the form $x_i(t)=\sum_{j=1}^{n}\sum_{l=1}^{L}b_{ij}(l)x_j(t-l) + \epsilon_i(t)$, where $L$ is the maximum lag, $b_{ji}(l)$ are linear regression coefficients, and $\epsilon_i(t)$ is white noise process [5]. After fitting the data the algorithm uses the AR coefficients to calculate the transfer functions involving $x_i$ and $x_j$ which are further used to estimate the frequency domain “causal effects” $(f_{i\rightarrow j},~ f_{j\rightarrow i}, f_{i j})$ which are then used to estimate the adjacency matrix of the causal graph.
>
> **References:**
>
> [1] Huang et al. "Causal discovery from heterogeneous/nonstationary data." JMLR. 2020.
>
> [2] Fujiwara et al. "Causal discovery for non-stationary non-linear time series data using just-in-time modeling." Conference on Causal Learning and Reasoning. PMLR, 2023.
>
> [3] Biswas, R., & Shlizerman, E. (2022). Statistical perspective on functional and causal neural connectomics: The Time-Aware PC algorithm. PLOS Computational Biology, 18(11), e1010653.
>
> [4] Rana, M. T., Veedu, M. S., & Salapaka, M. V. (2025). Uncovering the Influence Flow Model of Transistor Amplifiers, Its Reconstruction and Application. arXiv preprint arXiv:2508.04977.
>
> [5] Biswas, R., & Shlizerman, E. (2022). Statistical perspective on functional and causal neural connectomics: A comparative study. Frontiers in Systems Neuroscience, 16, 817962.
>
> [6] Kalisch, M., & Bühlman, P. (2007). Estimating high-dimensional directed acyclic graphs with the PC-algorithm. Journal of Machine Learning Research, 8(3).
>
> [7] Ding, M., Chen, Y., & Bressler, S. L. (2006). Granger causality: basic theory and application to neuroscience. Handbook of time series analysis: recent theoretical developments and applications, 437-460.

---

> > ### Comment · Reviewer_eWah · 2025-11-28
> > **Thank you**
> >
> > Thank you for the authors’ response and the new experiments, but my concerns remain.
> >
> > Although the response mentions a “real transistor circuit implemented on a printed circuit board,” the method description states that the data are obtained via “high-fidelity Cadence Virtuoso simulations.” Is my understanding correct that these results are simulation results rather than physical measurements from a hardware implementation? If they are indeed simulation results, I am still concerned about the claim that the proposed assumptions hold broadly in the “real world.” Simulations typically operate in idealized environments where mathematical constraints are easily satisfied, whereas physical systems introduce noise, delays, and heterogeneity that may violate your assumptions.
> >
> > Therefore, I strongly recommend that the authors provide concrete evidence (e.g., physical data or literature on non-ideal systems) supporting the validity of these assumptions in physical settings, or present a sensitivity analysis showing how the algorithm performs when these assumptions are partially violated.
> >
> > In addition, although the authors have explained the practical availability of sparsity constraints in power and circuit systems, the response does not substantively address the core concern about the algorithm’s sensitivity to sampling and frequency discretization.

---

> > > ### Author Response · Authors · 2025-12-03
> > >
> > > **Reviewer Comment:** "Thank you for the authors’ response and the new experiments, ... are partially violated."
> > >
> > > **Author response:**
> > >
> > > Thank you for responding to our rebuttal. We would like to clarify that our experiments did, in fact, include physical measurements obtained from actual hardware. In our first rebuttal to your original review, we had added two new sets of results: one where data was acquired from a **real hardware set up implemented on a printed circuit board (PCB)**, and another where data was acquired from high fidelity Cadence simulation.
> > >
> > > In the current new revised submission, we have added one more hardware-based experiment. Thus, currently we have two hardware-based case studies and one high-fidelity simulation-based case study. These results validate the efficacy and the applicability of our framework on real systems. We describe these experiments briefly in what follows.
> > >
> > > 1) *Bipolar Junction Transistor Based Circuits: Hardware validation:*
> > > The real-world results were presented for a **hardware** circuit containing bipolar junction transistors, capacitors, and resistors implemented on a printed circuit board. The circuit under test is shown in Fig. 2 of the updated article along with the PCB implementation. A picture of the PCB is shown in Fig. 2. Fig. 7 shows waveforms of the measured data set on an oscilloscope which was connected directly to the PCB board.
> > >
> > > 2) *MOSFET based Circuits: Hardware validation:*
> > > In addition to our earlier demonstrations on the real hardware, we have now added another experimental result on a different physical hardware set-up. The new hardware set-up consists of $2N7002E$ n-channel MOSFETs. The working principles of MOSFETs are different from that of BJTs, as a result their physical properties differ significantly. For example, MOSFETs typically have wider bandwidth and are faster in response time compared to BJTs. As a result, the MOSFET based circuits provide another avenue for validating the causal discovery algorithms. To perform the validation, we used a dataset containing $500000$ samples collected from a hardware set-up (See Fig. 3 in the updated article for a diagram of the hardware set-up) using the NI-cRio based data acquisition system described earlier (see Appendix C.9 for a detailed description of the data acquisition system). The comparative results are shown in the table below (These results are added to Section 6.4 in the article).
> > >
> > > | Algorithm    | F1 Score (%)   | CS (%) | TPR (%)  | 1-FPR (%) | Runtime (seconds) |
> > > | :---------------- |:-----: |:-----: |:------: | :--------: | :----------: |
> > > | Fisher-Z PC | 40  | 11.11 | 66.67 | 44.44 | 0.021 |
> > > | GC       | 85.71  | 88.89 | 100 | 66.67 | 0.0389 |
> > > | TPC     | 75  | 77.78  | 100 | 77.78 | 20.816 |
> > > | CD-NOD | 66.67  | 66.67  | 100  | 66.67  | 0.0323 |
> > > | FFT-WPC   | 100  | 100 | 100 | 100  | 0.0842 |
> > > | Wiener Phase | 100  | 100 | 100  | 100  | 0.0464 |
> > >
> > > Observe that the proposed method outperforms the SOTA methods in terms of both accuracy and speed in this data set also. This demonstrates the dominance of our proposed method for the tested real-world dynamical systems.
> > >
> > > The hardware-based case studies do suffer from many real-world imperfections which include
> > > * measurement noise introduced by the measurement equipment, i.e. NI-cRio, and AD8421 amplifier,
> > > * latent confounders due to noisy resistors, interference from power sources, and device temperature variations,
> > > * delays introduced by analog to digital converter of the NI-cRio data acquisition system,
> > > * heterogeneous noise sources including thermal noise, shot noise, and flicker noise introduced by the transistors,
> > > * parameter drift of the circuit components due to temperature and environmental factors leading to changed transfer function coupling between stages.
> > >
> > > The efficacy and applicability of our reconstruction is corroborated as can be assessed from the results.
> > >
> > > 3) *High-Fidelity Cadence Based Simulations:*
> > > The other set of results included validation of our algorithm on a dataset acquired from a large circuit simulated in high fidelity simulation software Cadence Virtuoso. It is to be noted that Cadence is the industry leader for integrated circuit design and simulations in Cadence are often considered highly accurate representation of actual circuits. We have not altered any of the models assumed by Cadence and therefore our simulations include all the non-idealities including non-linear behavior, noise and other imperfections that good simulation engines incorporate in their models. The simulation-based cadence experiment allows us to test the scalability of our approaches as well
> > >
> > > Please see **Section 6.4** and **Appendix C.9** for the detailed description of the hardware-based experiments and the description of the hardware set-up with diagrams. Also, see **Appendix C.10** for the details of the Cadence based results on the circuits. All the new results are highlighted in yellow.

---

> > > ### Author Response · Authors · 2025-12-03
> > >
> > > **Reviewer Comment:** "In addition, although the authors have explained the practical availability ... to sampling and frequency discretization"
> > >
> > > **Author response:**
> > >
> > > We acknowledge the reviewer’s concern regarding the effect of sampling and frequency discretization. To address your concern now we have added the process of determining how to sample the frequency interval $\Omega=[0,2\pi]$. As described in Section 2.5, the frequency interval $\Omega=[0,2\pi]$ is sampled at $N$ equally spaced points, where $N=2^a$ with $a$ being a natural number. To determine the value of $N$ to be used for the proposed algorithms, we perform reconstruction using different values of $N$ and then select the smallest value of $N$ that produces acceptable performance. The method is described below with an example.
> > >
> > > To demonstrate the method, we used the data generative model $a_{i,0}x_i(t)+a_{i,1}x_i(t-1)+a_{i,2}x_i(t-2) = \sum_{j\neq i}b_{ij}x_j(t-1) + e_i(t)$, where the coefficients $b_{ij}$ were chosen randomly from $[0.2,0.4]$, and $a_{i,k}$ were chosen such that the time-series remain bounded. The model was used for generating data for $25$ random DAGs each containing $10$ nodes and randomly generated edges. The datasets, each containing $128000$ samples, were used in the FFT-WPC and Wiener-Phase algorithms with the sparsity-based threshold tuning algorithm at different values of $N$. The average $F_1$ scores over the $25$ random DAGs are shown below (See Appendix C.11 for a plot of $F_1$ Score vs $N$).
> > >
> > > |F-1 Scores for FFT-WPC Algorithm:|||||||
> > > | :--------: |:-----: |:-----: |:------: | :--------: | :----------: |:----------: |
> > > |   Value of $N$  | 4  | 8 | 16  | 32 | 64 | 128 |
> > > | F1 Score (%) | 32.29  | 38.16 | 69.47 | 84.50 | 90.85 | 94.16 |
> > >
> > > |F-1 Scores for Wiener-Phase Algorithm:|||||||
> > > | :--------: |:-----: |:-----: |:------: | :--------: | :----------: |:----------: |
> > > |   Value of $N$  | 4  | 8 | 16  | 32 | 64 | 128 |
> > > | F1 Score (%) | 45.59  | 46.59 | 64.14 | 77.1 | 82.07 | 82.07 |
> > >
> > > It can be observed that the algorithms perform better as the value of $N$ increases. However, note that for $N \ge 32$ the improvement in algorithm performance with increasing $N$ is marginal. Although increasing the value of $N$ may increase the performance beyond $32$ for the presented example, it would also increase the computational burden as the complexity of the algorithms is $O(\log{N})$. Therefore, to maintain an acceptable tradeoff between complexity and performance we choose $N=32$ for further experiments.
> > >
> > > To further address your earlier comment regarding the thresholding process (“The approach hinges on heuristic thresholding of Wiener coefficients, ... and frequency discretization.”), we would also like to point out that the threshold-based approaches on the Wiener filters provide robustness to noise in real systems. As described in [8], in the presence of measurement noise in the time-series data the use of a threshold which is determined based on the power spectral density of the noise and the time-series guarantees accurate reconstruction of the moral graph.
> > >
> > > In addition, we would like to emphasize that heuristic threshold-based procedures are widely used in SOTA methods. For instance, the Granger causality implementation of the opensource python library *TimeAwarePC* applies a percentile threshold to the average of the $(f_{i\rightarrow j},~ f_{j\rightarrow i})$ when constructing the causal graph. Similarly, the TPC algorithm [3] employs an empirical threshold $\gamma$ on the relative frequency of edges across an ensemble of learned graphs to determine the final edge set. Therefore, we believe that the empirical threshold-tuning process used in our work should not be viewed as a limitation of our method alone, as analogous strategies are widely adopted in current state-of-the-art methods.
> > >
> > > **References**
> > >
> > > [1] Huang et al. "Causal discovery from heterogeneous/nonstationary data." JMLR. 2020.
> > >
> > > [2] Fujiwara et al. "Causal discovery for non-stationary non-linear time series data using just-in-time modeling." PMLR, 2023.
> > >
> > > [3] Biswas et al. "Statistical perspective on functional and causal neural connectomics: The Time-Aware PC algorithm." PLOS Computational Biology. 2022.
> > >
> > > [4] Rana et al. "Uncovering the Influence Flow Model of Transistor Amplifiers, Its Reconstruction and Application." arXiv, 2025.
> > >
> > > [5] Biswas et al. "Statistical perspective on functional and causal neural connectomics: A comparative study." Frontiers in Systems Neuroscience, 2022.
> > >
> > > [6] Kalisch et al. "Estimating high-dimensional directed acyclic graphs with the PC-algorithm." JMLR, 2007.
> > >
> > > [7] Ding et al. "Granger causality: basic theory and application to neuroscience. Handbook of time series analysis: recent theoretical developments and applications." 2006.
> > >
> > > [8] Materassi et al. "On the problem of reconstructing an unknown topology via locality properties of the wiener filter." IEEE TAC, 2012.

---

### Official Review · Reviewer_TpiQ · 2025-10-31

**Soundness:** 2
**Presentation:** 2
**Contribution:** 2
**Rating:** 4
**Confidence:** 4

**Summary:**

This paper investigates causal structure recovery in linear dynamical systems using Wiener Filter (WF) based approaches. The authors compare time-domain (TD) and frequency-domain (FD) methods for computing WF coefficients and propose two algorithms: Wiener-PC (extending PC algorithm with WF-based conditional independence tests) and Wiener-Phase (exploiting phase properties of WF for efficient MEG reconstruction). The work provides theoretical analysis including concentration bounds and sample complexity for both approaches, demonstrating computational advantages of FD methods.

**Strengths:**

1: The Wiener-Phase algorithm cleverly exploits the phase properties of frequency-domain WF coefficients to efficiently identify colliders and reconstruct MEGs, which has no analogue in time-domain approaches.

2: The evaluation includes synthetic data with controlled network structures, scalability analysis up to 50 nodes, and validation on real-world river-runoff dataset, showing consistent advantages of FD approaches.

**Weaknesses:**

1: The Wiener-Phase algorithm relies on two strong assumptions (phase alignment of incoming edges and non-zero imaginary components) that significantly limit its applicability. While the authors claim these apply to "a large class of networks," the conditions are quite specific and may not hold in many real-world scenarios.

2: The comparison primarily focuses on traditional static methods (Fisher-Z, chi-square tests) and Granger causality, missing more recent state-of-the-art causal discovery methods for time series data.

3: I'm unclear on where the distinction between static and dynamic approaches is specifically presented in your algorithm, and your specific contribution is unclear.

4: While the paper shows results for up to 50 nodes, the computational complexity analysis suggests the methods may not scale well to truly large networks due to the combinatorial nature of PC-based approaches. Please increase the credibility and workload.

**Questions:**

1: How can practitioners verify whether Assumptions 1 and 2 hold for their specific applications? What happens to algorithm performance when these assumptions are violated beyond the limited robustness study provided?

2: Given the O(n^(q+3)) complexity of Wiener-PC, how does the method perform on networks with hundreds or thousands of nodes? What are the practical limits of the approach?

3: While Lemma A.1 suggests any frequency can be used, how does the choice of frequency affect performance in practice, especially in noisy or finite-sample settings?

---

> ### Author Response · Authors · 2025-11-22
> **Part-1**
>
> **Reviewer Comment:** The Wiener-Phase algorithm cleverly exploits t ... to 50 nodes, and validation on real-world river-runoff dataset, showing consistent advantages of FD approaches."
>
> **Author Response:**
>
> Thank you for pointing out the strengths of the article. To further improve the article, we have now added another baseline (CD-NOD) for comparison. Moreover, we have added new experimental results showing the efficacy of the Winer-Phase and FFT-WPC algorithm over SOTA methods. The new demonstration involves datasets acquired circuits consisting of transistors which were implemented in physical hardware, thus further establishing the applicability of the proposed methods in real-world application. Please see Section 6.4 of the article for the new experimental details.
>
> **Reviewer Comment:** "The Wiener-Phase algorithm relies on two strong assumptions (phase alignment of incoming edges and non-zero imaginary components) that significantly limit its applicability. While the authors claim these apply to "a large class of networks," the conditions are quite specific and may not hold in many real-world scenarios."
>
> **Author Response:**
>
> As acknowledged in the article, the assumptions may seem restrictive. However, the assumptions hold in many real-world applications such as, consensus dynamics, thermal RC networks, power grid dynamics, linearized chemical reactions, and many electronic circuits as listed in Appendix C.6. To further support our claim, we have now added results in the article showing the efficacy of the Wiener-Phase algorithm in real transistor-based circuits implemented on printed circuit boards. A summary of the results is provided below.
>
> The authors of [4] have shown that the relationships among the output voltages of different amplifier stages can be expressed in the modeling framework of our article here. Therefore, to apply our proposed method to electronic circuits we acquired datasets for a multistage transistor amplifier from high fidelity simulation in cadence virtuoso. The circuit consists of $13$ transistors and $9$ measurement points with resistive and capacitive elements connecting the various stages. For a detailed diagram of the circuit please see the updated article. A set of $9$ time-series measurements containing $799999$ samples each was obtained. For the given circuit we compared the reconstruction results of our proposed methods with that of the base line algorithms. The results are reported in the table below:
>
> | Algorithm    | F1 Score (%)   | CS (%) | TPR (%)  | 1-FPR (%) | Runtime (seconds) |
> | :---------------- |:-----: |:-----: |:------: | :--------: | :----------: |
> | Fisher-Z PC | 37.84  | 37.74 | 70 | 67.74 | 0.1897 |
> | GC       | 80 | 91.94 | 100 | 91.94 | 0.0928 |
> | TPC     | 60.61  | 79.03 | 100 | 79.03 | 97.84 |
> | CD-NOD | 37.84 | 37.74  | 70  | 67.74  | 0.298 |
> | FFT-WPC   | 100 | 100  | 100  | 100  | 10.97  |
> | Wiener Phase | 90.91 | 96.8  | 100  | 96.8  | 0.153  |
>
> As described earlier, the dataset was generated from a circuit contains dynamics. Note that the Wiener-Phase algorithm produces almost perfect reconstruction for the circuit.
> To further validate our methods in real-world applications we compared the proposed method with the SOTA methods using data acquired from physical hardware setup. We acquired a dataset containing $4$ time-series voltage measurements with $500000$ samples from the authors of [4]. The circuit which was implemented on a printed circuit board consists of a multistage transistor amplifier built using MMBT2222ATT1G bipolar junction transistor from Onsemi, resistors, and capacitors. To measure and acquire the data the authors of [4] used instrumentation amplifiers and Ni-cRio data acquisition system. The table below summarized the comparative results of reconstruction.
>
> | Algorithm    | F1 Score (%)   | CS (%) | TPR (%)  | 1-FPR (%) | Runtime (seconds) |
> | :---------------- |:-----: |:-----: |:------: | :--------: | :----------: |
> | Fisher-Z PC | 40  | 0 | 100 | 0 | 0.019 |
> | GC       | 75  | 77.78 | 100 | 77.78 | 0.0391 |
> | TPC     | 60  | 55.56  | 100 | 55.56 | 23.641 |
> | CD-NOD | 40  | 0  | 100  | 0  | 0.0309 |
> | FFT-WPC   | 100  | 100 | 100 | 100  | 0.0798 |
> | Wiener Phase | 100  | 100 | 100  | 100  | 0.0419 |
>
> Observe that the proposed method outperforms the SOTA methods in terms of both accuracy and speed. Moreover, the Wiener-Phase algorithm produces perfect reconstruction of the generative graph of the circuit implemented in the hardware. This shows that indeed, Wiener-Phase algorithm is effective in such application domain.

---

> ### Author Response · Authors · 2025-11-22
> **Part-2**
>
> **Reviewer Comment:** "The comparison primarily focuses on traditional static methods (Fisher-Z, chi-square tests) and Granger causality, missing more recent state-of-the-art causal discovery methods for time series data."
>
> **Author Response:**
>
> We would like to emphasize that in our article we considered time-series that are dynamically related, that include, but not limited to, for example, systems with lag dependency. The Time-Aware PC (TPC) algorithm operates in dynamic systems and was demonstrated by the original authors [3] of being capable of handling dynamic dependencies in neuronal activity data. Moreover, the Granger causality (GC) algorithm is also applicable for time-series that are dynamically related, though GC is mostly effective when there is a definite non-zero “lag” in the dependencies.
>
> **Reviewer Comment:** "I'm unclear on where the distinction between static and dynamic approaches is specifically presented in your algorithm, and your specific contribution is unclear."
>
> **Author Response:**
>
> In our algorithms, we employ Winer filters to perform conditional independence tests. The Wiener filters given by $W_{i\cdot [j,z]}\[j\](\omega_k)$ identify the conditional independence test of dynamically related time series $x_i$ and $x_j$ given the set of time series $\\{x_r|r\in z\\}$. This makes our algorithms distinct from static causal discovery algorithms. The primary difference between the conventional static PC algorithm presented in  [6] and the proposed Wiener-PC algorithm is in the conditional independence metric. The methods such as partial correlation coefficient based conditional independence test used in [6] fail to capture lag dependencies. Since all algorithms proposed in the article rely on Wiener filters, they are applicable on dynamic systems.
>
> **Reviewer Comment:** "While the paper shows results for up to 50 nodes, the computational complexity analysis suggests the methods may not scale well to truly large networks due to the combinatorial nature of PC-based approaches. Please increase the credibility and workload."
>
> **Author Response:**
>
> We acknowledge the reviewer’s concern regarding the scalability of the approach since the PC algorithm in general has combinatorial explosion issues. We would like to point out that most causal discovery algorithms are combinatorial in nature and suffer from complexity explosion with number of nodes and lags. Our algorithms can excel in larger networks, where the current SOTA method may become intractable.
> - Compared to traditional time domain PC algorithms the proposed frequency domain PC algorithm provides improved scalability due to more computationally efficient conditional independence tests.
> - Most SOTA methods for dynamic systems, such as dynamic CD-NOD [1] and TPC [3], first estimate an unrolled DAG where the causal dependencies among different time segments of a time series are estimated. The unrolled DAG is then rolled back to obtain a causal graph. It can be inferred that in larger networks the unrolled DAGs can comprise a much larger number of nodes compared to the actual data generative graph, leading to an exacerbated combinatorial issue. Compared to such algorithms our proposed method does not involve estimating an unrolled graph, instead it estimates the actual MEG directly from data. Thus making the proposed algorithms more computationally efficient. Which means that the proposed methods can be applied to networks with large values of $n$ where current SOTA methods remain intractable.
> - Moreover, the Wiener-phase algorithm avoids combinatorial independence tests altogether, thus making it far more scalable to large networks. As demonstrated in Section 6.2 of the article, the Wiener PC algorithm is orders of magnitude faster compared to PC algorithms and SOTA methods such as TPC.
> - Note that our algorithm can handle infinite impulse response (IIR)models, whereas such models become intractable for the SOTA methods that rely on estimation of unfolded or unit graphs because of the complexity increase exponentially with the number of time indices. More precisely, for IIR convolution models the unrolled graph would have infinite numbers on nodes which will lead to intractability of such methods.

---

> ### Author Response · Authors · 2025-11-22
> **Part-3**
>
> **Reviewer Question:** "How can practitioners verify whether Assumptions 1 and 2 hold for their specific applications? What happens to algorithm performance when these assumptions are violated beyond the limited robustness study provided?"
>
> **Author Response:**
>
> To determine whether Assumptions 1 and 2 hold for the specific application one may rely on domain knowledge. For instance, if the practitioners are performing experiments in power networks, consensus dynamics networks, linearized chemical networks, and thermal networks then the assumptions are ensured to hold.
>
> However, if the reviewer is asking about how to validate whether the assumptions can be validated purely from data and theoretical results on the robustness of the algorithm then the answer would require a further involved study which we believe should be part of a future effort.
>
> **Reviewer Question:** "Given the O(n^(q+3)) complexity of Wiener-PC, how does the method perform on networks with hundreds or thousands of nodes? What are the practical limits of the approach?"
>
> **Author Response:**
>
> Given the complexity of the PC algorithm is $O(n^{q+3})$ they do suffer from combinatorial explosion. As a result, for hundreds or thousands of nodes the algorithm's runtime may be prolonged with a single core system. However, we can parallelize the algorithm to run on up to $n$ different parallel runs on a computer with multiple cores to reduce the algorithm run time. We would like to mention that the combinatorial nature of the PC algorithm was one of the primary motivations for developing the Wiener-Phase algorithm which does not have combinatorial explosion issues. The Wiener Phase algorithm  has $O(n^3)$ and thus is much more efficient compared to the PC algorithms. As a result, the Wiener Phase algorithm can excel in networks with hundreds or thousands of nodes with limited hardware capability.
>
> As with most causal discovery algorithms, in practice, constrained hardware resources can be a limiting factor for the PC based algorithm.
>
> **Reviewer Question:** "While Lemma A.1 suggests any frequency can be used, how does the choice of frequency affect performance in practice, especially in noisy or finite-sample settings?"
>
> **Author Response:**
>
> In noisy or finite sample settings the error in Wiener filter estimation may depend on the choice of frequency which may further affect the performance of the overall algorithms. However, different heuristics, such as choosing a random subset of frequency and averaging over them can improve robustness to such non-idealities. As mentioned in the article, we have observed the efficacy of such heuristics in the experiments presented in the article.
>
> **References:**
>
> [1] Huang et al. "Causal discovery from heterogeneous/nonstationary data." JMLR. 2020.
>
> [2] Fujiwara et al. "Causal discovery for non-stationary non-linear time series data using just-in-time modeling." Conference on Causal Learning and Reasoning. PMLR, 2023.
>
> [3] Biswas, R., & Shlizerman, E. (2022). Statistical perspective on functional and causal neural connectomics: The Time-Aware PC algorithm. PLOS Computational Biology, 18(11), e1010653.
>
> [4] Rana, M. T., Veedu, M. S., & Salapaka, M. V. (2025). Uncovering the Influence Flow Model of Transistor Amplifiers, Its Reconstruction and Application. arXiv preprint arXiv:2508.04977.
>
> [5] Biswas, R., & Shlizerman, E. (2022). Statistical perspective on functional and causal neural connectomics: A comparative study. Frontiers in Systems Neuroscience, 16, 817962.
>
> [6] Kalisch, M., & Bühlman, P. (2007). Estimating high-dimensional directed acyclic graphs with the PC-algorithm. Journal of Machine Learning Research, 8(3).
>
> [7] Ding, M., Chen, Y., & Bressler, S. L. (2006). Granger causality: basic theory and application to neuroscience. Handbook of time series analysis: recent theoretical developments and applications, 437-460.

---

### Official Review · Reviewer_oKCG · 2025-11-03

**Soundness:** 3
**Presentation:** 2
**Contribution:** 2
**Rating:** 4
**Confidence:** 4

**Summary:**

The authors tackle the problem of causal structure learning for systems where causal dependencies are non-static and dynamical in nature. They  provide a theoretical framework based on Wiener Filters (WFs) and Fast Fourier Transform (FFT) to introduce a novel method for deciphering non-static causal dependencies of dynamical systems from time-series data. They empirically validate their proposed approach on synthetic toy systems of 20 variables (nodes) and a real data (river-runoff) setting. This work provides a new perspective on causal structure learning of dynamical systems using frequency domain WFs.

**Strengths:**

- The authors introduce a novel, computationally efficient, method for causal structure recovery of non-static / non-stationary systems (systems with dynamic causal dependencies).
- The paper provides a novel way of thinking about causal structure learning of dynamical causal dependencies from time-series data; i.e. causal structure learning in the frequency domain. This insight leads to the aforementioned improvements in computational efficiency.

**Weaknesses:**

- Although in general this work is reasonably well written, and the authors did a good job at presenting the many nuanced details present in this work, this paper can still be hard to follow at times.
- The baselines in the empirical experiments seem to primarily be methods which operate in the static (stationary) setting. It would be useful to compare to some existing works that approach the problem of dynamic and non-stationary dependencies. To give some examples: [1, 2].
- Similarly, empirical experiments are quite limited. Although I believe this work presents comprehensive theoretical backing for their approach, which provides a reasonable degree of contribution already, the authors could consider some of the datasets/systems from [1, 2] to add to their  experiments section. Or perhaps, simulating data in some of the example systems the authors provide in Appendix C.6 could also be considered for addition experiments. I believe additional empirical experiment would help further strengthen this work.

**Questions:**

- In practical applications, how large does $N$ need to be to ensure reasonable recovery of the MEG?
- On lines 224-225: "For the settings with large number of samples (large $T$) or with longer delays (large $L$) improvements are significant". What are the practical settings where we would apply the proposed method and where there is sufficiently large $L, T$ such that it would yield competitive performance?
- Is the proposed approach practical for high dimensional systems (large $n$)?
- For the synthetic experiments in section 6, are the dynamical / non-stationary dependencies of the ground truth process/DAG random? Could the authors provide some additional clarification here?
- Lines 422-424: "Bounds on sparsity metrics such as average, maximum, and minimum degree are typically available ...". In practical applications and scientific applications, this is not necessarily know apriori, and thus, is not a reasonable assumption to make. Could the authors provide some examples supporting this claim?
- Could the authors provide further details for how the Granger baseline is implemented? Further, have the authors also considered a very simple correlation baselines? i.e. compute the correlation between samples across adjacent time-points and use this as the estimated DAG?
- The CS metrics seems like a weird choice. Why not consider something like average prevision / area under the average precision curve? You can compute this without thresholding, and it would provide a metric that captures some sort of notion of accuracy across "all thresholds".


Minor comments:

- I would define the acronym FFT (Fast Fourier Transform) earlier in the text (in fact I don't think its defined anywhere currently).
- Notation for $T$ slightly over-loaded. It is used to indicate transpose and in the number of time-series samples.

References:

[1] Huang et al. "Causal discovery from heterogeneous/nonstationary data." JMLR. 2020.

[2] Fujiwara et al. "Causal discovery for non-stationary non-linear time series data using just-in-time modeling." Conference on Causal Learning and Reasoning. PMLR, 2023.

---

> ### Author Response · Authors · 2025-11-22
> **Part-1**
>
> **Reviewer Comment:** "The authors tackle the problem ... systems using frequency domain WFs."
>
> **Author Response:**
>
> Thank you for your comprehensive summary of the article. We would also like to emphasize that the main focus of our article is on frequency domain methods and their efficacy over time domain approaches and there is no counterpart to a frequency domain analysis in the referenced articles. We have also illustrated the importance of using the phase in the frequency domain approach, which does not have a time domain counterpart. We now comment on each of your comments in detail.
>
> **Reviewer Comment:** "The authors introduce a novel, computationally efficient, method for causal structure recovery ... insight leads to the aforementioned improvements in computational efficiency.
> "
>
> **Author Response:**
>
> Thank you for summarizing the strengths of the article. We would also like to note that in addition to the computational efficiency of the frequency domain-based estimation approach, which you have pointed out, the article introduces a new algorithm, Wiener phase, which is significantly more efficient compared to conventional causal discovery algorithms. As mentioned in the article the new algorithm remains applicable to many practical systems. We have now presented experimental evidence of the applicability as per your request below.
>
> **Reviewer Comment:** " Although in general this work is reasonably well written, and the authors did a good job at presenting the many nuanced details present in this work, this paper can still be hard to follow at times."
>
> **Author Response:**
>
> Thank you for acknowledging that the article is reasonably well written. To further enhance the readability of the article we will further streamline the presentation emphasizing new definitions, use concise language whenever possible, and present more examples that illustrate the methods and their efficacy.

---

> ### Author Response · Authors · 2025-11-22
> **Part-2**
>
> **Reviewer Comment:** "The baselines in the empirical experiments seem to primarily be methods which operate in the static (stationary) setting. It would be useful to compare to some existing works that approach the problem of dynamic and non-stationary dependencies. To give some examples: [1, 2]."
>
> **Author Response:**
>
> We thank the reviewer for pointing us to the work in [1,2]. These works are indeed very relevant to the overall applications being targeted. An initial read of the articles and our methods indicate different approaches that likely have their own distinct advantages and disadvantages. From a quick read of the reference [1], we see that in [1] the causal dependencies could be nonlinear in the SCM. In our article we are assuming the underlying model to be linear, though dynamic. In our case the second order statistics are translation invariant (in the time variable). Here, the conditional distribution may change with time but they are translation invariant in the sense that statistics depend only on the time lag. A model of linear stochastic dynamic model as a generative model for the time series allows for a frequency domain approach which is the focus of article. Moreover, in reconstruction of topologies of dynamically related time-series where the network is unrolled over the time-index followed by employing techniques to assess dependencies on the unrolled network, to infer dependencies of the original network, the unrolled network dimensions can explode with the time window being unrolled. Moreover, in principle, the unrolling approach is not employable when the impact of one agent depends on the entire past of another, for example, when one timeseries is a IIR (infinite impulse response) filtered version of another, wherein it is difficult to estimate the horizon of the unrolling window. These issues are largely circumvented when employing a frequency domain approach.
>
> We would like to emphasize that tools and results of [1,2] and the comparative analysis bear closer scrutiny. In the discussion below we do provide some comparison. However, a deeper analysis is left for future study.
>
> We would like to emphasize that in our  article we considered time-series that are dynamically related, that include, but not limited to, for example, systems with lag dependency. The Time-Aware PC (TPC) algorithm operates in dynamic systems and was demonstrated by the original authors [3] of being capable of handling dynamic dependencies in neuronal activity data. Moreover, the Granger causality (GC) algorithm is also applicable for time-series that are dynamically related, though GC is mostly effective when there is a definite non-zero “lag” in the dependencies.
>
> Regarding the stationarity of the considered settings, the theoretical foundations in the article requires joint wide-sense stationarity of the time-series and not strict stationarity. In other words, the frequency domain based approaches require the auto correlation function $(E[x(t)x^T(t+\tau)])$ and cross correlation function $(E[x(t)y^T(t+\tau)])$ to be a function of the lag $(\tau)$  and are thus given by ($R_{xx}(\tau)$ and $R_{xy}(\tau)$ respectively that do not depend on the  time index $(t)$ (here, $x^T$ denotes the transpose of $x$ and $E$ is the expectation operator). The joint distribution is allowed to change in any other way. Practically, this assumption entails that transient influences are assumed to have died out in the dynamic response. We would like to emphasize that there is a significant class of applications where the time-scale of transients is short wherein our assumption of jointly wide sense stationarity holds. There is also the question of the causal structure itself changing over set of parameters (for example say time). We believe that our methods can be adapted for detection of online detection of such changes. Here we can envision that first, if for example the topology changes with time, then we can detect the change by employing methods in the article to windowed segments of data and subsequently detecting any changes in topology; here, our initial studies (not a part of this article) indicate that such an approach is computationally tractable. These extensions are the subject of future research.
>
>
>
> *Response is continued in the next comment due to character limit.

---

> ### Author Response · Authors · 2025-11-22
> **Part-3**
>
> **Reviewer Comment:** "The baselines in the empirical experiments seem to primarily be methods which operate in the static (stationary) setting. It would be useful to compare to some existing works that approach the problem of dynamic and non-stationary dependencies. To give some examples: [1, 2]."
>
> **Author Response:** *Continued from Part-2
>
> The real-world applications bear out that the assumptions made in the article employed while proving and modeling remain applicable. The river-runoff dataset contains average daily river runoff of $12$ measurement stations at different points in the Danube river system. The dataset contains measurement $4600$ samples recorded during $1960$ to $2009$. The river-runoff dataset is subject to fluctuations in weather conditions, including rainfall, temperature change, and snowmelt in the alpine regions. For example, according to “timeanddate.com” the average temperature of a city named Galati in the Danube basin during June is $73^\circ$ F, whereas average temperature in July and August is $77^\circ$ F with sudden changes in temperature over short time-scales is quite likely. However, the effectiveness of our methods here do support the conjecture that the transitionary effects remain manageable. We have also now added new real-world applications (on circuits that employ transistors)  where the time-series are dynamically related; we show that our methods remain effective (see response to your next comment).
>
> We have now performed comparisons with the baseline algorithm in reference provided by you. The CD-NOD algorithm of [1] was obtained from the causal-learn library then used on the river-runoff dataset which is a real-world dataset. We have compared the performance of our proposed frequency domain method with that of the CD-NOD algorithm as summarized in the table below. For the CD-NOD algorithm we used $\alpha=0.06$ and Fisher-Z based independence test.
>
> | Algorithm   | F1 Score (%)  | CS (%) | TPR (%)  | 1-FPR (%) | Runtime (seconds) |
> | :---------------- |:-----: |:-----: |:------: | :--------: | :----------: |
> | Fisher-Z PC | 52.63  | 64.32 | 76.92 | 87.4 | 0.526 |
> | GC       | 33  | 28.38  | 38.46 | 89.92 | 0.235 |
> | TPC     | 28.57  | 72 | 100  | 72 | 9244.53 |
> | CD-NOD | 54.05  | 65.16 | 76.92 | 88.24 | 0.1794 |
> | FFT-WPC   | 75  | 67.5  | 69.2   | 98.3 |  9.041  |
> | Wiener Phase | 48.3  | 46.29  | 53.85   | 92.44 | 0.02593 |
>
> The FFT-WPC was able to obtain comparable error performance with the best algorithm reported in [3], TPC, while run time was 1000 times faster. The FFT-WPC algorithm surpassed the CD-NOD algorithm in terms of CS.  Wiener phase was the fastest with a runtime of 26 milliseconds, but the CS was lower than FFT-WPC and TPC. However, Wiener-Phase was able to beat GC in CS. Note that the $F_1$ score also follows a similar trend. It is to be noted here that in the above table we reported the accuracy metrics from [3] for the TPC algorithm except for the $F_1$ score and run time. In our experiment we observed lower accuracy metrics for TPC possibly due to limited computation resources.

---

> ### Author Response · Authors · 2025-11-22
> **Part-4**
>
> **Reviewer Comment:** "Similarly, empirical experiments are quite limited. Although I believe this work presents comprehensive theoretical backing for their approach, which provides a reasonable degree of contribution already, the authors could consider some of the datasets/systems from [1, 2] to add to their experiments section. Or perhaps, simulating data in some of the example systems the authors provide in Appendix C.6 could also be considered for addition experiments. I believe additional empirical experiment would help further strengthen this work."
>
> **Author Response:**
>
> To address your concern regarding limited experiments, we have now added new results where we have applied the proposed methods of the article to circuits that employ transistors.
> The authors of [4] have shown that the relationships among the output voltages of different amplifier stages can be expressed in the modeling framework of our article here. Therefore, to apply our proposed method to electronic circuits we acquired datasets for a multistage transistor amplifier from high fidelity simulation in cadence virtuoso. The circuit consists of $13$ transistors and $9$ measurement points with resistive and capacitive elements connecting the various stages. For a detailed diagram of the circuit please see Section 6.4 of the updated article. A set of $9$ time-series measurements containing $799999$ samples each was obtained. For the given circuit we compared the reconstruction results of our proposed methods with that of the base line algorithms. The results are reported in the table below:
>
> | Algorithm    | F1 Score (%)   | CS (%) | TPR (%)  | 1-FPR (%) | Runtime (seconds) |
> | :---------------- |:-----: |:-----: |:------: | :--------: | :----------: |
> | Fisher-Z PC | 37.84  | 37.74 | 70 | 67.74 | 0.1897 |
> | GC       | 80 | 91.94 | 100 | 91.94 | 0.0928 |
> | TPC     | 60.61  | 79.03 | 100 | 79.03 | 97.84 |
> | CD-NOD | 37.84 | 37.74  | 70  | 67.74  | 0.298 |
> | FFT-WPC   | 100 | 100  | 100  | 100  | 10.97  |
> | Wiener Phase | 90.91 | 96.8  | 100  | 96.8  | 0.153  |
>
> As described earlier, the dataset was generated from a circuit contains dynamics. It can be observed that our proposed approach outperforms all other SOTA methods in such a scenario. Observe that the FFT-WPC produces perfect reconstruction. The second most accurate algorithm in terms of CS is the Wiener Phase algorithm which is also the fastest. Although Granger causality and TPC produce reasonably accurate results, they are surpassed by the proposed methods. Observe that the $F_1$ score also follows a similar trend, i.e., the FFT-WPC achieves the highest F1 score followed by Wiener-Phase, GC,  TPC, CD-NOD, and Fisher-Z PC in descending order.
> To further validate our methods in real-world applications we compared the proposed method with the SOTA methods using data acquired from physical hardware setup. We acquired a dataset containing $4$ time-series voltage measurements with $500000$ samples from the authors of [4]. The circuit which was implemented on a printed circuit board consists of a multistage transistor amplifier built using MMBT2222ATT1G bipolar junction transistor from Onsemi, resistors, and capacitors. To measure and acquire the data the authors of [4] used instrumentation amplifiers and Ni-cRio data acquisition system. The table below summarized the comparative results of reconstruction.
>
> | Algorithm    | F1 Score (%)   | CS (%) | TPR (%)  | 1-FPR (%) | Runtime (seconds) |
> | :---------------- |:-----: |:-----: |:------: | :--------: | :----------: |
> | Fisher-Z PC | 40  | 0 | 100 | 0 | 0.019 |
> | GC       | 75  | 77.78 | 100 | 77.78 | 0.0391 |
> | TPC     | 60  | 55.56  | 100 | 55.56 | 23.641 |
> | CD-NOD | 40  | 0  | 100  | 0  | 0.0309 |
> | FFT-WPC   | 100  | 100 | 100 | 100  | 0.0798 |
> | Wiener Phase | 100  | 100 | 100  | 100  | 0.0419 |
>
> Observe that the proposed method outperforms the SOTA methods in terms of both accuracy and speed. This demonstrates the dominance of our proposed method for the tested real-world dynamical systems.
>
> **Reviewer Question:** "In practical applications, how large does N need to be to ensure reasonable recovery of the MEG?
>
> **Author Response:**
>
> As described in the article, $N$ stands for the length of time series segments for calculating the FFT of time-series data. In practical applications this is typically determined by the largest lag dependency in the dataset. A value of $N$ larger than the largest lag would produce reasonable recovery given sufficient sample size. For instance, as described in [3], the physical time delay of interaction (inferred based on river velocity) between time series in the river run-off data is typically less than one day. As a result, we have observed a small $N=4$ to produce reasonable recovery of the MEG. However, in the case of electronic circuits, the lag dependencies were larger because of slow varying states. As a result, a larger value of $N=32$ was required to ensure accurate reconstruction.

---

> ### Author Response · Authors · 2025-11-22
> **Part-5**
>
> **Reviewer Question:** "On lines 224-225: "For the settings with large number of samples (large T) or with longer delays (large L) improvements are significant". What are the practical settings where we would apply the proposed method and where there is sufficiently large L, T  such that it would yield competitive performance?"
>
> **Author Response:**
>
> As described in the article $L$ stands for number of past and future samples considered for calculating the time domain Wiener filters and $T$ stands for the length of the time series. In other words, $L$ quantifies the lag dependencies in the time-series data and $T$ is the sample size. In many applications it may happen that the lag dependencies between the time series are large which would justify the use of a large $L$ and $T$. For instance, in transistor amplifiers and filters often different values of circuit parameters (resistance, capacitance, inductance) are used. The circuit parameters determine the time constants for circuits which can be often slow evolving. For example, use of a large resistor in combination with a capacitor leads to a slow evolving state of the capacitor and leads to a large lag which justifies use of large $L$ to yield competitive performance. Another instance where use of large $L$ would be required is a power system. In power systems large synchronous generators are bulky mechanical equipment whose states evolve on a slow time scale leading to large lag dependencies in the network.
>
> **Reviewer Question:** "Is the proposed approach practical for high dimensional systems (large n)?"
>
> **Author Response:**
>
> We acknowledge the reviewer’s concern regarding the scalability of the approach since the PC algorithm in general has combinatorial explosion issues. However, we would like to point out that our algorithms can excel in larger networks (large $n$), where the current SOTA method may become intractable.
>
> - Compared to traditional time domain PC algorithms the proposed frequency domain PC algorithm provides improved scalability due to more computationally efficient conditional independence tests.
>
> - Most SOTA methods for dynamic systems, such as dynamic CD-NOD [1] and TPC [3], first estimate an unrolled DAG where the causal dependencies among different time segments of a time series are estimated. The unrolled DAG is then rolled back to obtain a causal graph. It can be inferred that in larger networks the unrolled DAGs can comprise a much larger number of nodes compared to the actual data generative graph, leading to an exacerbated combinatorial issue. Compared to such algorithms our proposed method does not involve estimating an unrolled graph, instead it estimates the actual MEG directly from data. Thus, making the proposed algorithms more computationally efficient. Which means that the proposed methods can be applied to networks with large values of $n$ where current SOTA methods remain intractable.
>
> - Moreover, the Wiener-phase algorithm avoids combinatorial independence tests altogether, thus making it far more scalable to large networks. As demonstrated in Section 6.2 of the article, the Wiener PC algorithm is orders of magnitude faster compared to PC algorithms and SOTA methods such as TPC.
>
> - Note that our algorithm can handle infinite impulse response (IIR)models, whereas such models become intractable for the SOTA methods that rely on estimation of unfolded or unit graphs because of the complexity increase exponentially with the number of time indices. More precisely, for IIR convolution models the unrolled graph would have infinite numbers on nodes which will lead to intractability of such methods.

---

> ### Author Response · Authors · 2025-11-22
> **Part-6**
>
> **Reviewer Question:** "For the synthetic experiments in section 6, are the dynamical / non-stationary dependencies of the ground truth process/DAG random? Could the authors provide some additional clarification here?"
>
> **Author Response:**
>
> For the synthetic experiments of  the ground truth processes follow the dynamics dictated by the equation
> $a_{i,0}x_i(t)+a_{i,1}x_i(t-1)+a_{i,2}x_i(t-2) = \sum_{j\neq i}(b_{ij}x_j(t-1) + e_i(t)$,
> where the coefficients $b_{ij}$ were chosen randomly from $[0.2,0.4]$, and $a_{i,k}$ were chosen such that the time-series remain bounded.
>
> For the experiments in Section 6.1 we followed a two step process to generate the random data generative systems:
>
> Step-1: Generate a random DAG $G_k = (V,E_k)$, where $V:=\\{1,2,3,...,n\\}$ is the fixed number of nodes and $E_k$ is the randomly generated set of edges between the nodes in $V$. During the generation of the random edge set $E_k$ we first organize the nodes in $V$ in a chronological order. Then for each node $i\in V$ we select a random set of nodes from $\{1,2,...i-1\}$ as its parents. While choosing the parent set of node $i$, the maximum number of parents is limited to a predetermined value. In the experiments presented in section 6.1 the maximum possible number of parents for a node was chosen to be 2. Observe that although there can be only 2 incoming edges at a particular node due to this restriction, this does not limit the number of outgoing edges from a node.
>
> Step-2: Once a random DAG $G_k$ has been generated, we use its adjacency matrix to generate the coefficients of the lag dependencies in the model, i.e. the $b_{ij}$ coefficients. More precisely, $b_{ij}\sim U(0.2,0.4)$ if $(j,i)\in E_k$ and $b_{ij}=0$ if $(j,i)\notin E_k$.
>
> As mentioned in Section 6.1 of the article, we averaged the performance of our algorithm over $25$ networks. In each case we repeated the two steps above to generate the random DAGs.
> For the $20$-nodes and $50$-nodes experiment in Section 6.2,  the adjacency matrix was fixed according to the graphical structures shown in Appendix C3 and Appendix C.4 of the article. However, the coupling coefficients, $b_{ij}$, were chosen randomly from $[0.2,0.4]$ for both the network.
>
> **Reviewer Question:** " Lines 422-424: ... provide some examples supporting this claim?"
>
> **Author Response:**
>
> We would like to emphasize that sparsity bounds are available in many application domains. For example, in power system networks different bus/nodes are connected together through electric wires which are the transmission lines. In practice the design infrastructure of power networks imposes constraints on the connectivity of the nodes due to different reasons such as cost of design and operation and geographical location. As a result, a particular node in a power network can be connected to only certain geographical neighbors leading to constraints on the sparsity of the connectivity matrix. Moreover, this information is typically available to the power system operator.
>
> Another example is analog electronic circuits. In a typical electronic amplifier network the connectivity between different stages is upper bounded due to design constraints and complexity. Such constraints on design lead to sparsity patterns in the networks. Such information is typically known to designers and in many cases can be inferred from the datasheets of devices.
>
> **Reviewer Question:** "Could the authors provide further ... time-points and use this as the estimated DAG?"
>
> **Author Response:**
>
> The granger causality algorithm was obtained from the open source Python library known as *TimeAwarePC* by the authors of [3]. The *TimeAwarePC* library uses the granger causality implementation of the *nitime* python repository which follows the method described in [7] for estimation of the causal effects. The Granger causality algorithm fits the time-series to a multivariate auto regressive model of the form $x_i(t)=\sum_{j=1}^{n}\sum_{l=1}^{L}b_{ij}(l)x_j(t-l) + \epsilon_i(t)$, where $L$ is the maximum lag, $b_{ji}(l)$ are linear regression coefficients, and $\epsilon_i(t)$ is white noise process [5]. After fitting the data the algorithm uses the AR coefficients to calculate the transfer functions involving $x_i$ and $x_j$ which are further used to estimate the frequency domain “causal effects” $(f_{i\rightarrow j},~ f_{j\rightarrow i}, f_{i j})$ which are then used to estimate the adjacency matrix of the causal graph.
>
> We have considered baselines that are correlation based in the comparison section. The Fisher-Z based PC algorithm utilizes partial correlation-based metrics to determine conditional independence from data [6]. More precisely, the Fisher-Z test uses the estimated sample partial correlation $\rho_{i,j|z}$ and its Fisher’s z-transform to verify whether $x_i$ and $x_j$ are conditionally independent given $\\{x_r|r\in z\subset\\{1,2,...n\\}\setminus\\{i,j\\}\\}$.

---

> ### Author Response · Authors · 2025-11-22
> **Part-7**
>
> **Reviewer Question:** "The CS metrics seems like a weird choice. Why not consider something like average prevision / area under the average precision curve? You can compute this without thresholding, and it would provide a metric that captures some sort of notion of accuracy across "all thresholds"."
>
> **Author Response:**
>
> In the article we have considered TPR, FPR, and CS to show performance of the algorithms. TPR and FPR are metrics based on the confusion matrix, which is a typical choice in many classification problems. We have chosen these metrics because many of the baseline experiments such as TPC and GC reported this metric as a performance measure [3]. To have a fair comparison with what is presented in pertinent literature we chose the CS, TPR and FPR metrics in the comparison section. To address your concern we have now added the F1 score which is a widely used metric in evaluating performance of binary classification. The score was calculated as $F1 =\frac{2TP}{2TP+FP+FN}$.
>
> We note that metrics such as area under ROC curve and precision-recall curve are typically used for algorithms that generate probability scores for classification instead of $0$ or $1$ labels. However, the causal discovery algorithms identify $0$ or $1$ entries of an adjacency matrix of a graph where such metrics are not typical choices. Hence, we used the TPR, FPR and now the F1 score metric.
>
> **Reviewer Comment:** "I would define the acronym FFT (Fast Fourier Transform) earlier in the text (in fact I don't think its defined anywhere currently)."
>
> **Author Response:**
>
> Thank you for pointing this out. We shall define the acronym explicitly when it is introduced first in the updated article.
>
> **Reviewer Comment:** "Notation for T slightly over-loaded. It is used to indicate transpose and in the number of time-series samples."
>
> **Author Response:**
>
> Thank you for pointing this out. To improve clarity, we shall change the notation of transpose to $\top$ instead of $T$ to avoid overloading the notations.
>
> **References:**
>
> [1] Huang et al. "Causal discovery from heterogeneous/nonstationary data." JMLR. 2020.
>
> [2] Fujiwara et al. "Causal discovery for non-stationary non-linear time series data using just-in-time modeling." Conference on Causal Learning and Reasoning. PMLR, 2023.
>
> [3] Biswas, R., & Shlizerman, E. (2022). Statistical perspective on functional and causal neural connectomics: The Time-Aware PC algorithm. PLOS Computational Biology, 18(11), e1010653.
>
> [4] Rana, M. T., Veedu, M. S., & Salapaka, M. V. (2025). Uncovering the Influence Flow Model of Transistor Amplifiers, Its Reconstruction and Application. arXiv preprint arXiv:2508.04977.
>
> [5] Biswas, R., & Shlizerman, E. (2022). Statistical perspective on functional and causal neural connectomics: A comparative study. Frontiers in Systems Neuroscience, 16, 817962.
>
> [6] Kalisch, M., & Bühlman, P. (2007). Estimating high-dimensional directed acyclic graphs with the PC-algorithm. Journal of Machine Learning Research, 8(3).
>
> [7] Ding, M., Chen, Y., & Bressler, S. L. (2006). Granger causality: basic theory and application to neuroscience. Handbook of time series analysis: recent theoretical developments and applications, 437-460.

---

> > ### Comment · Reviewer_oKCG · 2025-11-27
> >
> > Thank you for your very detailed rebuttal! The authors have addressed all my concerns and questions. With this, I will raise my score. It is unclear to me (or hard to see, since there is not text in different color) if the authors have updated the manuscript with their clarifications. If not done so already, I recommend the authors incorporate the changes.

---

> > > ### Author Response · Authors · 2025-12-03
> > >
> > > Thank you for raising your score. We are glad to know that we were able to address all your concerns.
> > >
> > > Moreover, we have now highlighted the updated/added text in the new revision of the manuscript. Below is a summary of the changes made in the revised version of the article:
> > >
> > > * We have added Section 6.4 containing two new sets of results on experiments on circuits that contain transistors, resistors, and capacitors.
> > >
> > >      * The first set of results were presented for a circuit containing bipolar junction transistors, capacitors, and resistors implemented on a printed circuit board; therefore, the related results are obtained via real hardware and not simulation based. The circuit under test is shown in Fig. 2 of the updated article along with the printed circuit board (PCB) implementation. The hardware-based results validate the efficacy of our methods under real-world imperfections, such as measurement noise, latent confounder due to interference from power sources and device temperature variations, delay due to analog-to-digital conversion, heterogeneous noise sources, and parameter drift.
> > >
> > >     * The second set of results are also experimental results on a different physical hardware set-up. The new hardware set-up consists of $2N7002E$ n-channel MOSFETs. The working principles of MOSFETs are different from that of BJTs, as a result their physical properties differ significantly. For example, MOSFETs typically have wider bandwidth and are faster in response time compared to BJTs. As a result, the MOSFET based circuits provide another avenue for validating the causal discovery algorithms.
> > >
> > >     * We have added Table 2 where we compared our algorithm with the State of the Art (SOTA) methods on the real hardware-based datasets to demonstrate the efficacy of our methods.
> > >
> > > * In Appendix C.9, we have described the hardware setup used for performing the data acquisition on the transistor-based circuits implemented on a printed circuit board. A diagram showing waveforms of the recorded measurements on an oscilloscope is also included.
> > >
> > > * In Appendix C.10, we have compared the performance of the proposed algorithms with the SOTA methods on a larger circuit simulated in Cadence Virtuoso which is the industry leader for high fidelity circuit simulators. Here we made no modifications to the high-fidelity models employed by Cadence and therefore the simulation-based results come with high confidence.
> > >
> > > * In Appendix C.11, we have presented the effects of frequency discretization on the performance of the proposed algorithms in experiments. A method to determine the value of $N$ is also given.

---

### Meta-Review · Area_Chair_okHg · 2026-01-05

**Summary:**

The paper compares time- versus frequency-domain estimation of multivariate Wiener filters for causal structure learning under a linear dynamical influence model, and proposes a method for learning non-static causal dependencies from dynamical systems. Reviewers generally viewed the work as providing new insights into causal structure learning in the frequency domain, along with clear gains in computational efficiency and empirical performance.

While initial concerns were raised regarding e.g. empirical studies, real-world validation, strict assumptions, heuristic technique, and scalability, most of them were addressed during the discussion period.

**Reviewer Concerns:**

Concerns addressed by rebuttals:
- To address concerns about empirical studies (e.g., lack of comparison with approaches in dynamic/non-stationary setting and choice of metrics), the authors added comparisons with CD-NOD and provided additional metrics like F1 score.
- The empirical validation are quite limited. The authors addressed this by providing additional experiments on real-world hardware datasets and high fidelity simulation.
- Reviewers pointed out that the assumptions required can be strict. These are **partially** addressed by the authors' discussion of real-world examples, as well as additional experiments on two real-world hardware datasets and one from high fidelity simulation.
- The method relies on heuristic thresholding (rather than statistically calibrated procedure) and can be sensitive to sampling and frequency discretization. The authors clarified the approach for selecting the frequency sampling parameter and provided further analysis for its effect on performance.
- The proposed method may not be scalable. The authors provided discussion to show how the method is more scalable than existing ones.

**Reviewer Scores:**

Reviewer oKCG decided to increase their score (most probably to 6), although it is not reflected on the review. Reviewers TpiQ and eWah may have increased their scores to 6 if they had been able to participate fully in the discussion.

---

### Decision · Program_Chairs · 2026-01-26

Accept (Poster)